# A Systematic Literature Review on Diabetic Retinopathy Using an Artificial Intelligence Approach

Pooja Bidwai [1,*] , Shilpa Gite [1,*], Kishore Pahuja [2] and Ketan Kotecha [1,*]

1 Department of Artificial Intelligence and Machine Learning, Symbiosis Centre for Applied Artificial Intelligence (SCAAI), Symbiosis Institute of Technology, Symbiosis International (Deemed University) (SIU), Lavale, Pune 412115, India
2 Natasha Eye Care and Research Centre, Pimple Saudagar, Pune 411027, India
* Correspondence: pooja.bidwai.phd2021@sitpune.edu.in (P.B.); shilpa.gite@sitpune.edu.in (S.G.); director@sitpune.edu.in (K.K.)

**Abstract:** Diabetic retinopathy occurs due to long-term diabetes with changing blood glucose levels and has become the most common cause of vision loss worldwide. It has become a severe problem among the working-age group that needs to be solved early to avoid vision loss in the future. Artificial intelligence-based technologies have been utilized to detect and grade diabetic retinopathy at the initial level. Early detection allows for proper treatment and, as a result, eyesight complications can be avoided. The in-depth analysis now details the various methods for diagnosing diabetic retinopathy using blood vessels, microaneurysms, exudates, macula, optic discs, and hemorrhages. In most trials, fundus images of the retina are used, which are taken using a fundus camera. This survey discusses the basics of diabetes, its prevalence, complications, and artificial intelligence approaches to deal with the early detection and classification of diabetic retinopathy. The research also discusses artificial intelligence-based techniques such as machine learning and deep learning. New research fields such as transfer learning using generative adversarial networks, domain adaptation, multitask learning, and explainable artificial intelligence in diabetic retinopathy are also considered. A list of existing datasets, screening systems, performance measurements, biomarkers in diabetic retinopathy, potential issues, and challenges faced in ophthalmology, followed by the future scope conclusion, is discussed. To the author, no other literature has analyzed recent state-of-the-art techniques considering the PRISMA approach and artificial intelligence as the core.

**Keywords:** artificial intelligence; diabetic retinopathy; domain adaptation; explainable AI; fundus; optical coherence tomography (OCT)

## 1. Introduction

Ophthalmology is a medical specialty that focuses on the scientific research of diseases and diagnosing and treating various eye disorders. Ophthalmologists used to diagnose eye problems manually, which took a long time [1]. Diabetes is a long-term illness that interferes with our body's average capacity to digest food. Most of our foods are broken down into glucose and enter our bloodstream. When blood sugar levels rise, our pancreas is pushed to secrete insulin. Insulin is the element that permits blood glucose to enter our body's cells and then be used as food. Whenever a person develops diabetes, the body either does not create enough insulin or does not utilize it that well. There is more blood glucose when insufficient insulin or cells stop producing insulin. Complications of diabetes [1] include diabetic retinopathy (eye damage), neuropathy (nerve damage), nephropathy (kidney disease), cardiomyopathy (heart problems), gastroparesis, skin problems, etc. [1–4]. In primarily elderly populations, eye problems are the leading cause of blindness.

Furthermore, according to a World Health Organization (WHO) report, as the world's population ages, patients suffering from ocular disorders are predicted to increase [5,6]. As a result, there is a lot of interest in applying artificial intelligence (AI) to improve ocular

treatment while simultaneously cutting healthcare costs, especially when telemedicine is included [7,8]. Compared to the number of medical facilities accessible, the ratio of people suffering from eye disease is vast [9]. The most common causes of visual impairment are diabetic retinopathy, macular degeneration because of growing older, and glaucoma, a disease that affects the eyes. Cataracts are a type of aberration, and macular edema is a type of edema that affects the retina. Neovascularization of the choroids (CNV), retinal detachment, refractive errors, amblyopia, and strabismus are some of the retinal problems that may lead to a poor visual prognosis.

## 1.1. Applications of AI in Retina Images

There are three primary use case situations in retina image applications: classifying, segmentation, and predictions, which are shown in Figure 1.

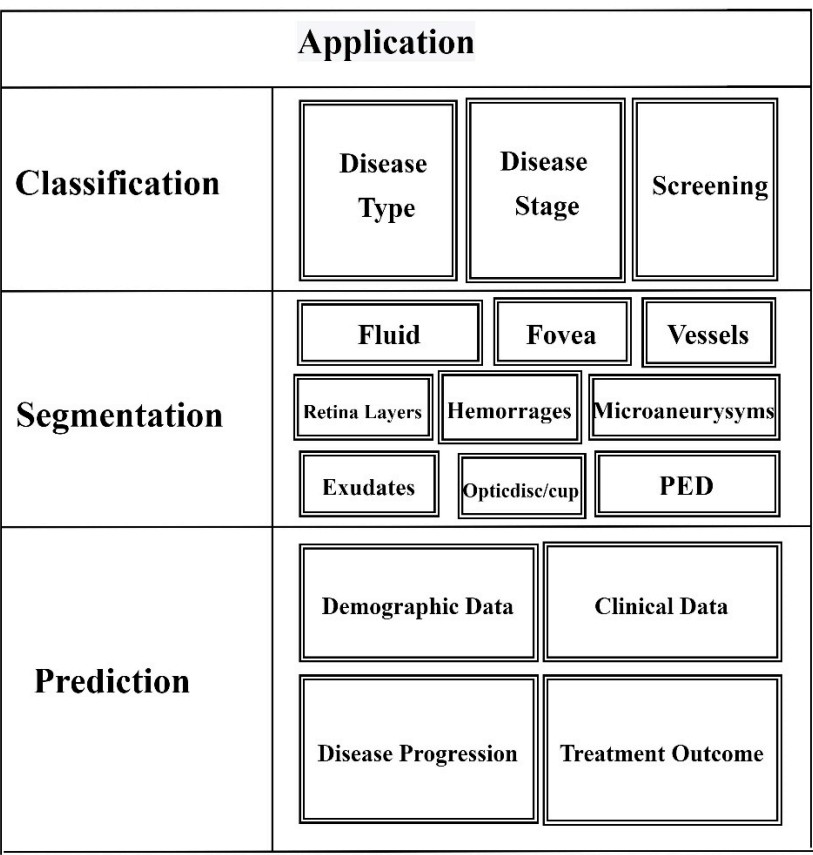

**Figure 1.** Applications of AI in retina imaging.

- Classification: Categorization cases are commonly used in binary or multi-class retinal image analysis, such as automated screenings or detecting of the stage of disease or type. ML and DL methods are applicable here based on the level of understandability required or the quantity of the dataset provided.
- Segmentation: The fundamental goal of segmentation-based approaches is to subdivide the objects in a picture. The primary purpose of all these techniques is to investigate morphological features or retrieve a meaningful pattern or feature of relevance from a snapshot, such as borders in 2D or 3D imaging. Segmentation of pigment epithelial detachment (PED) is used to diagnose chorioretinal diseases.
- Prediction: Most predicted situations are alarmed with illness development, future treatment outcomes based on an image, etc. The prediction approach can also be used to depict the local retention region.

### 1.2. Diabetic Retinopathy (DR)

DR is a sight-threatening illness resulting from damage to the retinal vessels, and it is increasing at an exponential rate. Diabetic problems are widespread and, as a result, this condition affects blood vessels, which then affect the retina's light-sensitive components. The primary cause of this disease's progression is a deficiency of oxygen delivered to the retina [10]. Persons with a long poor glycemic control are much more susceptible to causing it; whether an individual is type 1 or type 2 diabetic, the illness rises as they age [11,12]. DR is a secret illness that only emerges through its latter stages when therapy is impossible. Frequent retinal scanning is necessary for diabetic individuals to effectively treat DR at an earlier time to avert disability [13]. Automated screenings are essential to save manual tasks because the cost of such a method is high [14]. Additionally, because most of the population is above 45 years of age, a non-invasive procedure would be beneficial. According to the researchers, fundus imaging is a comfortable and non-invasive technology optometrists use to determine DR severity levels. Parameters such as microaneurysms (MAs), hemorrhagic (HEMs), exudates (EXs), and cotton wool spots are examined (CWS) for the detection of DR. There is a need for technology that allows a non-technical person to take a picture on [15–18] a mobile phone and email it to ophthalmologists, who can then subsequently advise their patients by looking at the picture on their phone.

Early detection can overcome DR. In the current circumstances, AI, along with related approaches, such as ML and DL in computer science, has proved to be a powerful tool for detecting complicated patterns in ocular illnesses. Computerized DR detection [19] techniques save cash and effort and are much more effective than traditional diagnoses [20]. Figure 2 shows the difference between a standard vision image and DR.

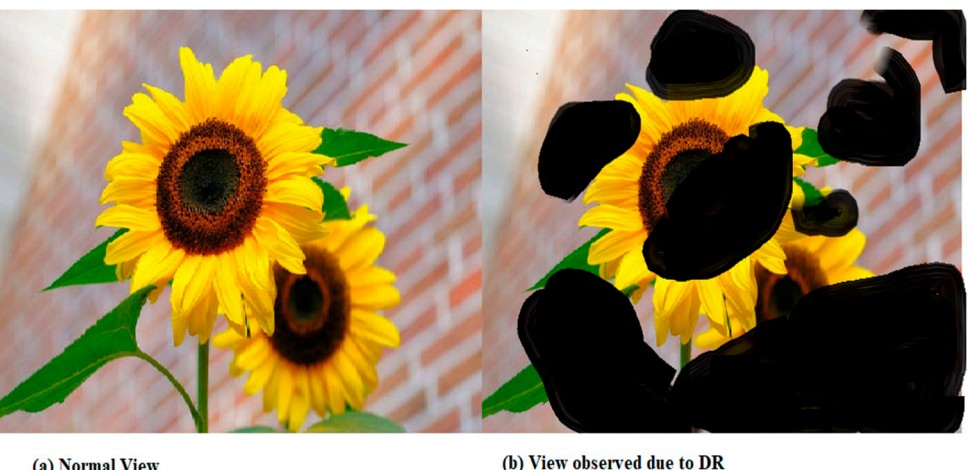

(a) Normal View    (b) View observed due to DR

**Figure 2.** Normal vision and DR vision.

An occurrence of various sorts of lesions on an eye can be used to diagnose a DR image, as shown in Figure 3. Lesions of microaneurysms (MA), hemorrhages (HM), and hard and soft exudates (EX) are shown [2–4].

Microaneurysms (MA): Microaneurysms (MA) in the fundus picture is an early clinical symptom of DR, causing retinal dysfunction due to blood/fluid leaking on the retina [21,22]. It appears as small red spots on the retina [23]. They may be encircled by a yellow lipid ring or hard exudates. They are surrounded by sharp borders and would be less than 125 μm. Endothelial dysfunction in these microaneurysms can result in leakage and retinal edema, leading to visual loss. The criterion for correctly detecting MAs is fluorescein angiography (FA), whose shapes fall into various categories such as focal bulge, saccular, fusiform, mixed, pedunculated, and irregular [24].

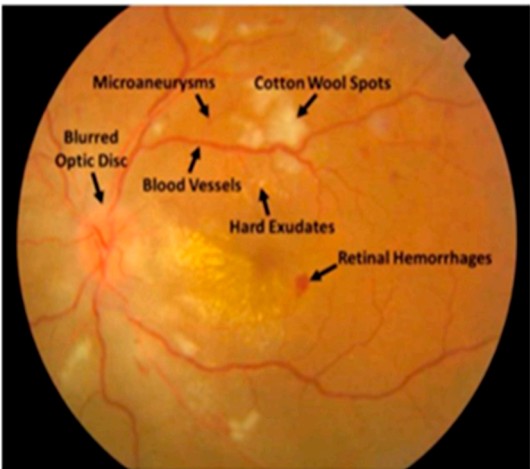

**Figure 3.** Representation of a fundus image with the lesion annotations.

- Hemorrhages (HM) appear as patches on the retina which can be 125 μm in diameter with an uneven edge. Its two categories are flames (superficial HM) and blot (deep HM) [23].
- Hard exudates: Hard exudates, which typically can be seen as bright yellow areas on the eye, are caused by hemolysis. These were also found in the eye's coastal parts and had clear boundaries.
- Soft exudates: White spots on the eye generated from nerve fiber swelling are called soft exudates (cotton wool). These are ovular or circular. Soft or hard secretions constitute white lesions, whereas MA and HM were red growths (EX). A sample image of various stages of DR is provided in Figure 4. DR is classified as non-proliferative DR (NPDR) and proliferative DR (PDR). Further, NPDR is classified as mild, moderate, and severe, as shown in Figure 5.

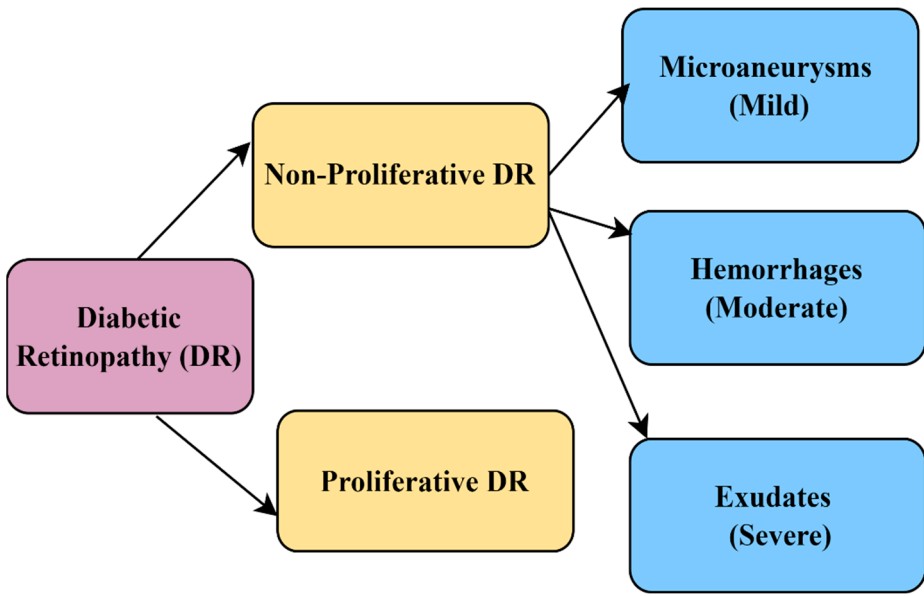

**Figure 4.** Stages of DR [25].

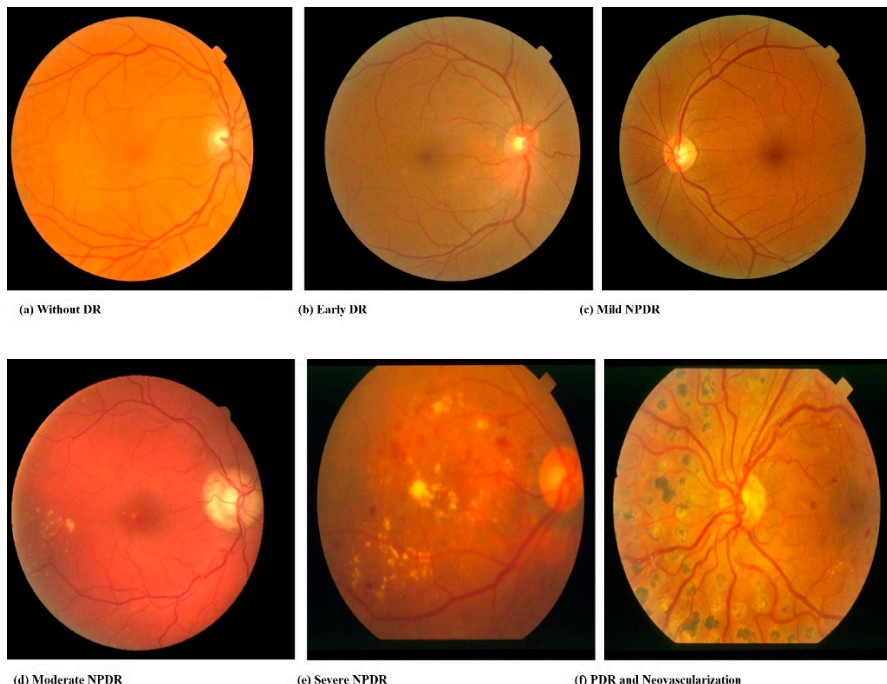

**Figure 5.** Classification of diabetic retinopathy.

Later phases, as opposed to earlier phases such as mild, moderate, and NPDR, are much more extreme versions in which additional frail vessels emerge. Such blood veins bleed blood into the ocular, hurting a human's sight. Fuzzified eyesight, decreased color recognition, dark spots, or threads swimming across our eyesight, changing vision, etc., are all symptoms of this disorder [10]. The typical situation involves the absence of DR lesions and fluid leakage from arteries.

Mainly in the event of milder NPDR, at minimum, one MA with/without HEMs, exudates, plus CWS, could be seen. MAs, HEMs, exudates, and CWS, less serious than severe NPDR, are known as moderate NPDR. Severe NPDR is characterized by more than a MA and HEM in four quadrants, venous beading in two quadrants, and intra-retinal and mild vascular abnormalities in one quadrant. PDR is identified by the massive size of the optic disc, cup, blood vessels, or pre-retinal vitreous, with DR abnormalities [26,27]. Furthermore, DR is the most significant cause of visual impairment in most developing and industrialized countries, especially among working individuals [28,29]. Furthermore, the manual DR evaluation of a specialist is arbitrary and firmly based on our technical experience. As a result, computerized technologies are urgently needed to quantify DR on a more extensive dataset reliably and reduce inter- and intra-reader variability [30,31].

### 1.3. Evolution of DR Using AI

In the United States, Europe, and Asia, it is estimated that around one-third of patients with diabetes (34.6%) have DR to some degree [32]. It is also worth noting that one out of every ten people (10.2%) has vision-threatening DR [33]. Vision loss caused by DR has been proven preventable with timely treatment. Enormous work has been performed in this field.

Additionally, there are various methods for detecting DR. Multiple DL-based fully automated DR monitoring methodologies have been introduced in the latest years. This section discusses some of the present research performed on DR with AI. The evolution of DR with artificial intelligence is shown in Figure 6. There are pioneering works carried out in this evolution. One of the works, presented by Gardner et al. [34] in 1996, stated that neural networks can detect diabetic features in fundus images and compare the network against an ophthalmologist screening a set of fundus images. Cree et al. [35] proved that computer vision techniques were suitable to detect microaneurysms. Their experiments

relied on simple morphological and thresholding techniques using eight features among pixel area and total pixel intensity measured on each candidate. The proposed method achieved similar results to those obtained by clinicians and proved that automated detection can be used for diagnostic purposes. Franklin [36] proposed a novel retinal imaging method that segments the blood vessels automatically from retinal images. This method segments each image pixel as vessel or non-vessel, which, in turn, is used for automatic recognition of the vasculature in retinal images. Retinal blood vessels were identified by means of a multilayer perceptron neural network, for which the inputs were derived from the Gabor and moment invariant-based features. Later, the methods evolved to detect not only microaneurysms in the fundus, but also the stage of diabetic retinopathy using ML and DL approaches [37]. A detailed literature review is discussed in Sections 3.1 and 3.2.

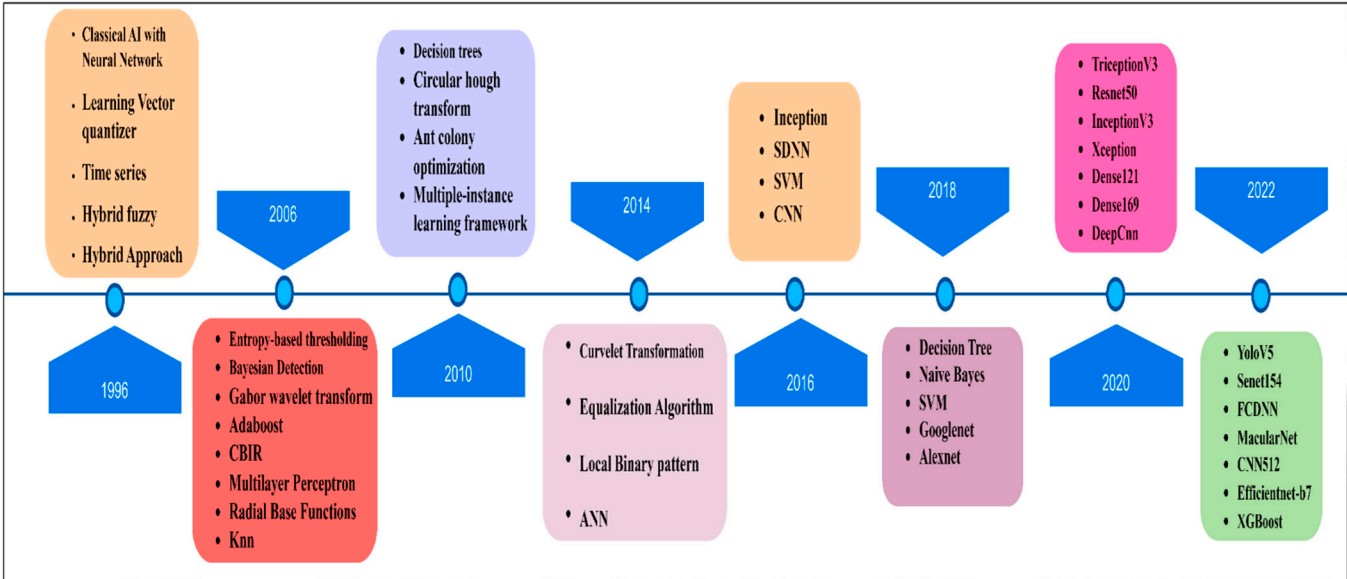

**Figure 6.** Evolution of DR with artificial intelligence.

### 1.4. Prior Research

According to our understanding, only a few systematic literature review (SLR) publications on DR using AI Technology are available. Jahangir R. [15] covers five main areas: data, preprocessing method, methods used in ML- and DL-based approaches, and performance evaluations used in DR detecting tactics. Contrast enhancement paired with green stream removal yielded the best classification results in picture preprocessing approaches. Among several features considered for DR detection, structure, texture, and statistical features were determined to be the most discriminative [25]. Compared to other ML classifiers, the artificial neural network has proven to be the most effective. The convolutional neural networks outperformed other DL networks in DL.

Pragathi P [38] suggested an integrated ML approach that incorporates support vector machines (SVMs), principal component analysis (PCA), and moth-flame optimization approaches. The DR dataset is first subjected to the ML algorithm's decision tree (DT), SVM, random forest (RF), and naïve Bayes (NB). The SVM algorithm outperforms the others, with an average performance of 76.96%. The moth-flame optimization technique is combined with SVM and PCA to increase the performance of ML systems. This proposed approach outperforms all previous ML algorithms with an average performance of 85.61%, and the accurate categorization of class labels is accomplished. Wei Zhang [39] discusses DeepDR. It uses transfer learning and ensemble learning to detect DR in fundus images. It consists of cutting-edge neural network works built from common convolutional neural networks and custom standard deep neural networks. DeepDR is developed by creating a high-quality dataset of DR medical images and then having clinical ophthalmologists label

it. The primary objective of this review would be to study the existing scholarly articles and related conclusions on the specified study topic. Table 1 lists the study topics that will aid in focusing on this SLR. To the author's knowledge, this study ensures detailed SLR, which is the first to cover methodologies for the detection of DR based on the AI methodology for robust and reliable detection.

**Table 1.** Highlights of earlier research related to DR.

| Ref No | Objectives and Topic | Discussions | Type |
|---|---|---|---|
| [15] | Datasets, picture preparation methods, ML-based methods, DL-based strategies, and evaluation metrics are presented as five components of DR screening methodologies. | Did not follow the PRISMA approach. Studies that were released between January 2013 and March 2018 are considered in this study. | Review |
| [39] | It discusses DeepDR, an automated DR identification, and grading system. DeepDR uses transfer learning and ensemble learning to detect the presence and severity of DR in fundus images. | Did not follow the PRISMA approach. Experiment results indicate the importance and effectiveness of the ideal number and combinations of component classifiers in model performance. | Review |
| [38] | It discusses an integrated ML approach that incorporates support vector machines (SVMs), principal component analysis (PCA), and moth flame optimization approaches for DR. | Did not follow the PRISMA approach. Utilizing the PCA technique to reduce the dimensions has had a detrimental impact on the performance of the majority of ML algorithms. | Review |
| [40] | It presents the latest DL algorithms used in DR detection, highlighting the contributions and challenges of recent research papers. | Did not follow the PRISMA approach. Robust deep-learning methods must be developed to give satisfactory performance in cross-database evaluation, i.e., trained with one dataset and tested with another. | Review |
| [41] | It presents a comprehensive survey of automated eye diseases detection systems using available datasets, techniques of image preprocessing, and deep learning models. | Studies that did not follow the PRISMA approach are considered from January 2016 to June 2021. | Review |

The major highlights of this paper are:

1. Datastores in the discipline of DR detection are accessible online, as well as the existence of DR datasets.
2. An exhaustive survey of widely used ML and DL methodologies for DR detection is discussed.
3. Feature extraction and classification techniques used in DR are discussed.
4. Future research concepts such as domain adaptation, multitask learning, and explainable AI in DR detection are discussed

### 1.5. Motivation

Working in the discipline of ophthalmology is primarily motivated by a concern for human health. The eye is among the essential sensory inputs since it receives data from the world and then transmits it to the central nervous system. Our mind subsequently converts the information collected by the eyes to usable information. As a result, our eyes are crucial senses which provide us with knowledge of the world, enabling us to try new knowledge, participate in artistic processes, and generate lovely memories. There is plenty of work to be performed in today's industrialized world utilizing various modern personal digital assistants, laptops, smartphones, etc. Additionally, because of the influence of COVID-19 over the last two years, most people who work from home utilize various internet platforms. As a result of all these factors, most people are experiencing vision problems. Various disorders such as obesity, cardiovascular disease, hypertension, strokes, and depression are more prevalent in persons with vision problems [37]. They are also more likely to stumble, become hurt, or become depressed.

As per recent publications, surveys [42], and clinical information, many people have indeed been detected with DR, AMD, cataracts, glaucoma, CNV, drusen, corneal scarring, and a variety of other eye ailments [33,38]. AI-related techniques have been used to diagnose eye-related disorders, and there is still a lot of AI potential to be discovered. Because AI and related approaches would fundamentally alter vision care, it would be a good possibility for the healthcare business as the element of AI is only beginning to be unveiled. As a result, there is a lot of interest in using AI to improve ophthalmologic treatment while lowering healthcare costs. This review further highlights a variety of allied methodologies and datasets to keep up with the field of ophthalmology's rapid growth. This publication aims to assist young researchers in a greater understanding of visual retinal problems and to work in optics to develop a self-contained platform.

### 1.6. Research Goals

According to the earlier studies and their outcomes, this research compares DR with AI and, accordingly, research questions are proposed to obtain a comprehensive survey of DR detection using AI. Table 2 shows the grouped survey items to make this study more comprehensive by enlisting research questions considered during SLR.

**Table 2.** Research questions.

| RQ. No. | Research Question | Objective/Discussion |
|---|---|---|
| 1 | What are the most common artificial intelligence-based methods for DR detection? | It assists in determining the most relevant artificial intelligence algorithms for DR diagnosis applications nowadays. |
| 2 | What are the various Features Extraction Techniques for DR? | List various feature extraction techniques used for DR. |
| 3 | What are the relevant datasets for DR? | Discovers several publicly available datasets that may be used as benchmarks to compare and assess the performance of various methodologies, as well as gives new researchers a head start. |
| 4 | What are the various evaluation measures used for DR detection? | The most used standards and metrics for DR detection are reviewed. |
| 5 | What are the potential solutions for a robust and reliable DR detection system? | It makes it easier to find significant research areas to be studied. |

Specific perspectives must be available to help academics generate innovative thinking by evaluating related studies. The first research question examines previously published work and the most common AI-based DR detection methods. The goal of research question 2 is to create a list of all feature extraction techniques used in DR [43]. Research question 3 will outline relevant datasets for DR exploration. Research question 4 will look at a few prominent evaluation measures in DR utilizing AI methodologies. In research question 5, existing effective methodologies' limitations and future directions constraints are listed.

### 1.7. Contribution of the Study

Our systematic literature review made the following contributions:

1.  To exploring available data sets which have been used for detecting DR.
2.  To investigate artificial intelligence strategies that have been employed in the literature for DR detection.
3.  To explore feature extraction and classification.
4.  To study multiple assessment metrics to analyze DR detection and categorization.
5.  To highlight the scope of future research, concepts such as domain adaptation, multi-task learning, and explainable AI in DR detection techniques used in DR.

This survey includes the research methodology for artificial intelligence-based ophthalmic analysis and the study's contribution in Section 2. The literature on techniques-based ophthalmology analysis is discussed in length in Section 3. In addition, a comparative examination of approaches and findings achieved by various ophthalmology researchers has been reviewed. In Section 4, a comparative examination looked at the potential difficulties and implications of related procedures in the optics domain. Section 5 clarifies the overview by outlining clinical applications, future research, and avenues for studying various diseases, diagnoses, and treatments for eye disorders. Section 6 discusses performance measures, Section 7 introduces biomarkers in DR, and Section 8 highlights research challenges with future research directions. The analysis is arranged to categorize and evaluate existing publications to encompass the study's breadth. The first step in defining the study topics is that the inclusion ratio of prior work could be precisely calculated. Some views can help scientists develop new innovative ideas by examining related spadework. Figure 7 demonstrates the organization of the Systematic Literature Review.

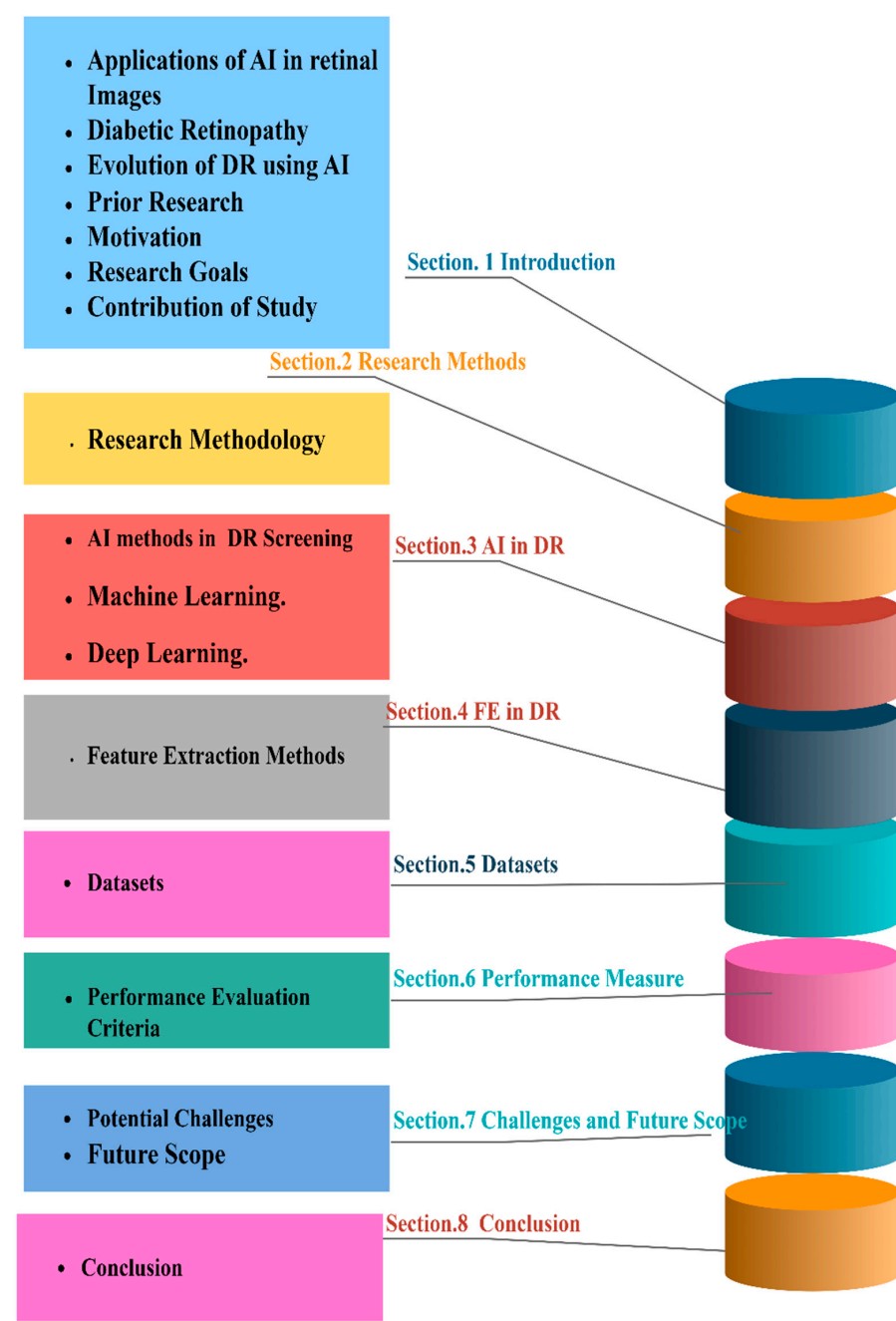

**Figure 7.** Demonstrates the organization of our SLR into various portions.

Table 2 provides a list of research questions used in SLR. The identification of information sources is the second phase in our SLR. Databases such as Scopus and Web of Science were used to find related papers. Some general and specific keywords are included in Table 3 that can be used to create search queries for finding research publications.

**Table 3.** Direct and indirect keywords were used.

| Fundamental Keyword | "Diabetic Retinopathy" | | | |
|---|---|---|---|---|
| Direct Keyword | "Artificial Intelligence" | "Machine Learning" | "Deep Learning" | |
| Indirect Keyword | "Ophthalmology" | "Fundus Images" | "DR Stages" | "OCT" |

## 2. Research Mechanism of Study

A descriptive analysis was carried out using the eligible study items for a structured literature review and meta-analysis (PRISMA) method. PRISMA comprises a list of protocols for longitudinal studies and other data-driven concepts, which includes writing and formatting guidelines. A three-step technique is employed to conduct a systematic review: the creation of research questions, online databases, and guidelines for accepting and discarding scientific papers of such research analysis processes are presented in the following pages.

The analysis is arranged systematically to categorize and evaluate existing publications to encompass the study's breadth. The first step in defining the study topics is that the inclusion ratio of prior work could be precisely calculated. Some views can help scientists develop new innovative ideas by examining related spadework. Table 2 provides a list of the relevant studies used in SLR. The identification of information sources is the second phase in our SLR. Scopus and Web of Science were used to find related papers. Some general and specific keywords are included in Table 3 that can be used to create search queries for finding research publications. The third step is to develop techniques for evaluating the technical and scientific documents. These results uncovered identified publications related to our condition. The proposed approach is divided into two sections: (I) choose queries that are used to search for and collect all relevant data using Boolean AND/OR, and (II) use Boolean operators AND/OR to find out search keywords from survey questions to make a note of topics. Table 4 lists the web searches used for this research. Figure 8 gives relevance of paper distribution based on (a) Data Source, (b) Year, and (c) Document Type.

**Table 4.** Search terms employed.

| Database | Query | Initial Outcome |
|---|---|---|
| Scopus | (Diabetic AND Retinopathy AND Artificial AND Intelligence AND Machine AND Learning AND Deep AND Learning) | 149 |
| Web of Science | | 79 |

*Paradigms for Inclusion and Exclusion*

A set of study protocols for selecting and excluding factors for rejections of scientific studies was developed to pick relevant scholarly articles for the literature review (Table 5). Three inclusion criteria phrases are employed in the screening procedure,

(a) Insignificant scientific studies were weeded out depending on the info and terms found in study summaries. Summaries of scientific papers that fulfilled a minimum of 40% of an IC are maintained for other stages.

(b) Full-text screening: Articles with summaries that only address limited elements of the keyword search are rejected if they do not reference or connect to a particular keyword in Table 3.

(c) Step of quality assurance: The leftover scientific studies were subject to something such as a qualitative review, and those that did not meet any of the eligibility principles were eliminated.

- RC1: Recommendations and results must be included in research articles.
- RC2: Scientific data must be included in scientific papers to support their conclusions.
- RC3: The aims and findings of the research must be expressed.
- RC4: For scientific studies, citations must be proper and adequate.

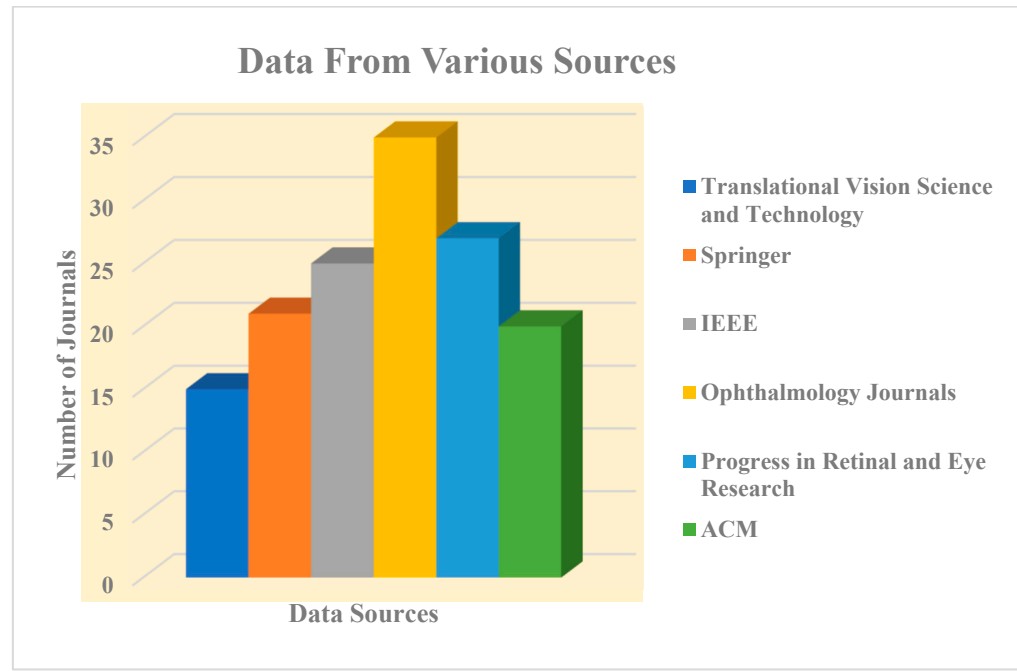

(**a**)

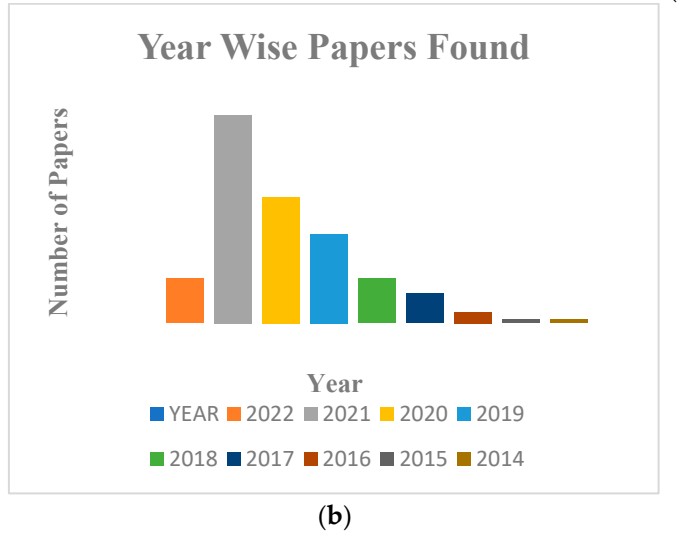

(**b**)

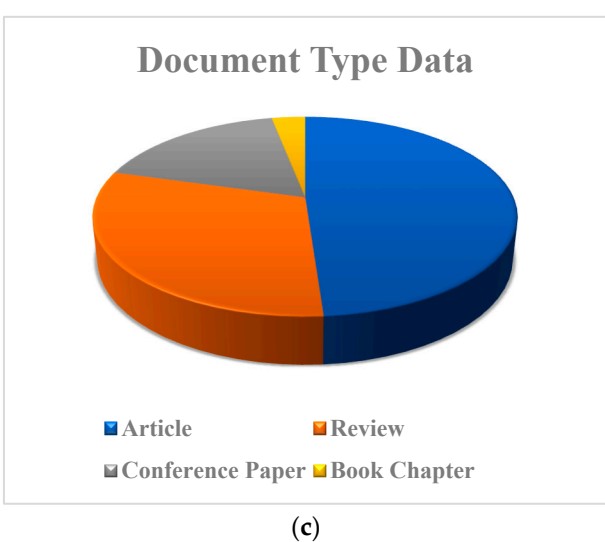

(**c**)

**Figure 8.** Relevant paper distribution based on (**a**) Data Source, (**b**) Year, and (**c**) Document Type.

**Table 5.** Inclusion and exclusion criteria considered.

| Inclusion Criteria |
| --- |
| Rather than reviews or survey pieces, scientific papers should be primary research papers. |
| Scholarly articles that appeared between 2014 and April 2022. |
| Query terms must be included in the titles, abstracts, or whole body of peer-reviewed publications. |
| Articles that address at least one research question. |
| The developed solution should aim at resolving issues with diabetic retinopathy detection using AI. |
| **Exclusion Criteria** |
| Articles that are written in languages other than English. |
| Studies published that are identical. |
| Complete scientific papers are not always available. |
| Research papers that are not related to diabetic retinopathy using AI. |

## 3. RQ1 Artificial Intelligence for DR Detection

The first half of this section discusses artificial intelligence-based DR detection tools, ML techniques, DL techniques, and transfer learning (Table 6). DR is a leading primary cause of preventable blindness worldwide, and artificial intelligence technologies can help with early detection, monitoring, and treatment. The lack of ground truth criteria in retinal exam datasets concerns supervised artificial intelligence algorithms that require high-quality exams. Various groups have used AI methods, such as ML and DL, to construct automated DR detection tools [44]. Such AI-based technology has been demonstrated to reduce prices, improve detection ability, and increase universal care for DR screening. Latest DL studies in optics suggest that it could substantially change human evaluators while retaining a good exactness.

**Table 6.** Major automated diabetic retinopathy tools.

| Software | Sample Size | Only DR OR Controls | Device | Grading/ Mechanism | Limitation | Software Mechanism | Used | Accuracy |
|---|---|---|---|---|---|---|---|---|
| Bosch [44] | 1128 | DR with age of 18+. | Bosch Mobile Eye Care fundus camera. Single field non-mydriatic. | ETDRS. | In some of the eyes diagnosed as normal, the other eye may have had early evidence. Further, while the study notes the findings of DR, it would be useful to know how accurate this software is for individual lesions, such as exudates, microaneurysms, and macular edema. | CNN-based AI software. | For DR screening in India. | Sensitivity—91%. Specificity—96%. Positive predicted value (PPV)—94%. Negative predictive value (NPV)—95%. |
| Retmarker DR [45] | 45,148 | Screening diabetic patients. | Used non-mydriatic cameras. Canon CR6-45NM with a Sony DXC-950P 3CCD color video camera other cameras, such as Nidek AFC-330 and CSO Cobra, have been used temporarily. | Coimbra Ophthalmology. Reading Centre (CORC). | The short duration of the study (2 years) and the lack of more detailed information on systemic parameters, such as lipid stratification. | Feature-based ML algorithms. | Used in local DR screening in Portugal, Aveiro, Coimbra, Leiria, Viseu, Castelo Branco, and Cova da Beira. | R0—71.5%, RL—22.7%, M—2.2%, RP—0.1%, NC—3.5%. Human grading burden reduction of 48.42%. |
| Eye Art [46] | 78,685 | A cross-sectional diagnostic study of individuals with diabetes. | Two-field undilated fundus photograph. Two-field retinal CFP images (one disc-centered and one macula-centered) were taken for each eye (Canon CR-2 AF or Canon CR-2 Plus AF; Canon USA Inc.). | ETDRS. | A limitation of the study is that optical coherence tomography was not used to determine clinically significant macular edema. Color fundus photographs CFP is known to be an accurate, sufficient, and widely accepted clinical reference standard, including by the FDA. | AI Algo. | Used in Canada for detection of both mtmDR and vtDR without physician assistance. | Sensitivity—91.7%. Specificity—91.5%. |

**Table 6.** *Cont.*

| Software | Sample Size | Only DR OR Controls | Device | Grading/ Mechanism | Limitation | Software Mechanism | Used | Accuracy |
|---|---|---|---|---|---|---|---|---|
| Retinalyze [47] | 260 | Retrospective cross-sectional study of diabetic patients attending routine. | Mydriatic 60° fundus photography on 35-mm color transparency film was used, with a single fovea-centered field fundus camera (CF-60UV; Canon Europa NV, Amstelveen, The Netherlands) set. | Routine grading was based on a visual examination of slide-mounted transparencies. Reference grading was performed with specific emphasis on achieving high sensitivity. | Commercially unavailable for a long time until reintroduced into its web-based form with DL improvements. | Deep learning based. | Used in Europe to a greater extent. | Sensitivity 93.1% and specificity 71.6%. |
| Singapore SERI-NUS [48] | 76,370 SIDRP between 2010 and 2013 (SIDRP 2010–2013) | With diabetes. | FundusVue, Canon, Topcon, and Carl Zeiss nonmydriatic. | Grading was completed by a certified ophthalmologist and retina specialist. | Identification of diabetic macular edema from fundus photographs may not identify all cases appropriately without clinical examination and optical coherence tomography. | Using a deep learning system. | Singapore. | Sensitivity 90.5% and specificity 91.6%. AUC—0.936 |
| Google [49] | 128,175 Aravind Eye Hospital, Sankara Nethralaya, and Narayana Nethralaya | Macula-centered retinal fundus images were retrospectively obtained from EyePACS in the United States and three eye hospitals in India among patients presenting for diabetic retinopathy screening. | Two sets of 9963 Eyepacs images from Centervue DRS, Optovue iCam, Canon CR1/DGi/CR2, and Topcon NW using 45° FOV and 40% acquired with pupil dilation. Images from a 1748-Messidor-2 from a Topcon TRC NW6 nonmydriatic camera and 45° FOV with 44% pupil dilation. | DR severity (none, mild, moderate, severe, or proliferative) was graded according to the International Clinical Diabetic Retinopathy scale. | Further research is necessary to determine the feasibility of applying this algorithm in the clinical setting and to determine whether the use of the algorithm could lead to improved care and outcomes compared with current ophthalmologic assessment. | CNN based. Inception-v3 architecture. | Used in North Carolina to a greater extent. | Sensitivity—97.5%. Specificity—93.4%. |

| Software | Sample Size | Only DR OR Controls | Device | Grading/ Mechanism | Limitation | Software Mechanism | Used | Accuracy |
|---|---|---|---|---|---|---|---|---|
| IDx-DR [50] | 900 | With no history of DR. | Widefield stereoscopic photography mydriatic. | FPRC Wisconsin Fundus Photograph Reading Center, and ETDRS. | The prevalence of referable retinopathy in this population is small, which limits the comparison to other populations with higher disease prevalence. | AI-based logistic regression model. | Dutch diabetic Care system-1410. | Sensitivity—87.2%. Specificity—90.7%. |
| Comprehensive Artificial Intelligence Retinal Expert (CARE)system [51] | 443 subjects (848 eyes) | Previously diagnosed diabetic patients. | One-field color fundus photography (CFP) (macula-centered with a 50◦ field of vision) was taken for both eyes using a nonmydriatic fundus camera (RetiCam 3100, China) by three trained ophthalmologists in dark rooms. | International Clinical Diabetic Retinopathy (ICDR) classification criteria. | This technique has drawbacks when it comes to detecting severe PDR and DME. (1) Poor imaging results from fundus such as ghost images and fuzzy lesions, in leukoplakia, lens opacity, and tiny pupils. Cases create difficulty in AI identification. (2) The difference in the results was caused by the study's insufficient sample size. (3) Some lesions were overlooked during the 50-degree fundus photography focused on the macula. | AI-based. | Chinese community health care centers. | Sensitivity—75.19%. Specificity 93.99%. |

### 3.1. Machine Learning Techniques in DR Detection

Several ML techniques are developed for DR classification. They are discussed below and a diagrammatic explanation is given in Figure 9

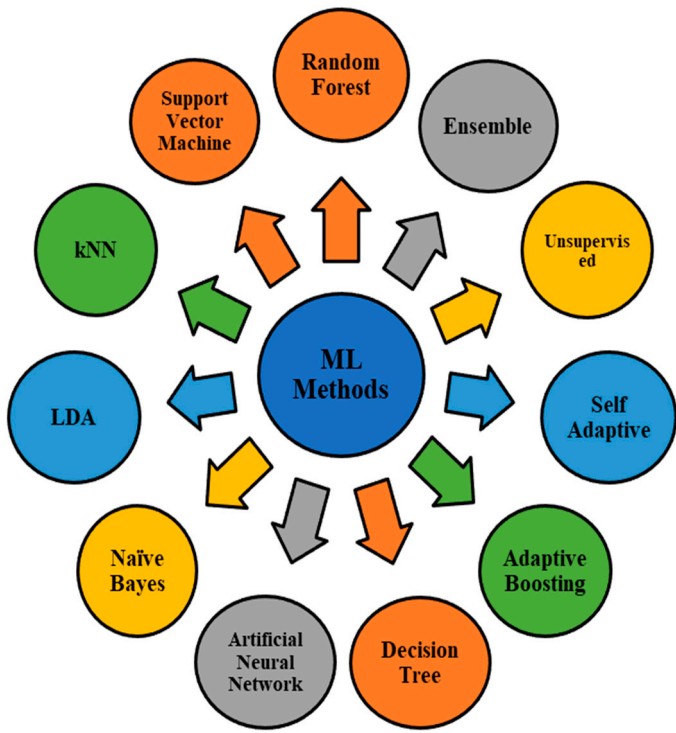

**Figure 9.** Machine learning methods.

(a)   Linear Discrimination Analysis: The local linear discriminating study forms the most extensively utilized classifications and dimension reduction methods. It can be used to discriminate between multiple classes. It finds a projection to a line, allowing samples from various classes to be separated [15]. In the critical ML investigations, LLDA was utilized only once. Wu and Xin [52] used the LLDA algorithm to detect microaneurysms and compared the results with the SVM and k-NN in the ROC dataset. The authors found that the LLDA algorithm failed to perform, and the SVM gave good results for both the LLDA and k-NN, in terms of accuracy.

(b)   Decision Trees: A fundamental tool for solving classification tasks. Its structure is similar to a tree and hierarchical, where an internal node represents a test on an attribute, a branch represents a test outcome, and a terminal node carries a class label. A root node is a topmost node in a tree. It is used to represent decisions in decision analysis. One of the benefits of a DT is that it does not necessitate much data preparation. One downside of a DT is that it can occasionally result in overly complicated DTs, often known as overfitting. With DIARETDB0 and DIARETDB1 datasets, Rahim and Jayne [53] detected microaneurysms from retinopathy images using SVM and k-NN.90% of the total photos were used for training, while the remaining 10% were used to test the classification algorithms.

(c)   Support Vector Machines: SVMs (support vector machines) is a CNN model for categorizing data. This generates a binary classifier around the datasets example (support vectors and hyperplane). Accordingly, the A+ and A− categories denote the nearest distances toward the positively and negatively extreme examples. A hyperplane is a plane that separates the A+ and A− levels, featuring A+ on one side of the hyperplane and A− on another [15]. In various studies, researchers successfully employed SVM techniques to classify distinct DR conditions, including [54–58]. Furthermore, the authors claim that the SVM improves classification performance. SVM, SCG-BPN, and GRN exudates within retinal photos have been detected and classified using

techniques by Vanithamani and Renee Christina [59]. The DIARETDB1 dataset, consisting of 40 training and 40 testing images, was used. The experimental findings revealed that the SVM algorithm outperformed the SCG-BPN and GRN algorithms in classification performance.

The naïve Bayes (NB) classifier is probabilistically based. It works with numerical data to build a model. It takes significantly less amount of numeric data to predict classification.

(d)    Naïve Bayes: As a result, it is a quick and straightforward categorization algorithm. Wang and Tang [60] examined three classification systems for microaneurysm detection: NB, k-NN, and SVM. Tests were conducted on the private and public datasets. k-NN outperformed in its research compared to the NB method.

(e)    K-Nearest Neighbor: One of the most basic and straightforward ML techniques is the k-nearest neighbor (k-NN) methodology. It classifies items in a feature set using the training set's nearest instances. The character "k" shows the cluster count utilized by the classifiers to build its prediction. Among the 40 publications about ML, the k-NN algorithm was employed in numerous investigations [15]. The k-NN method was utilized by Nijalingappa and Sandeep [61] to classify DR into severity stages. They employed 169 photos from Messidor and DIARETDB1 datasets and a unique dataset [53] in their research. They used 119 photos to train their ML method and 50 photographs to test it. The classification results produced by k-NN are satisfactory.

(f)    Random Forest: Random forest is an effective and successful ML classification method. It forms decision trees in forests (DT). The projections will be much more accurate if enough bushes are in the forests. Every tree casts a judgment about categorizing a novel method depending on its characteristics, and the models are then stored with the tree's name. The category with the best scores is chosen by forest. To put it differently, an RF classification technique is comparable to the bagging technique. A subset of the training dataset is formed in RF, and a DT is created for each subset. In the testing set, every input sequence is classified by all the DTs, as well as the forest chooses the one with the best scores [15]. The RF classifier was only used once in the experiments that were chosen. The RF classifier was used by Xiao and Yu [62] to detect bleeding in retinal images. They used 35 photos from a unique dataset and 55 images from DIARETDB1. They employed 70% of the photos for training the ML network, and the remaining 30% were used for testing and classification with the RF technique. The RF algorithm acquired good sensitivity, according to the findings of the experiments.

(g)    Artificial Neural Networks: This classifier comprises three layers: input nodes, hidden nodes, and an output vector. There seem to be numerous nodes in the input and hidden nodes, but only one in the output nodes. A neuron is a type of activating unit in a neural network. Patterns are sent from the input nodes layer, which does the actual processing. The node with hidden units is allocated random weights. The output node is equipped with a hidden layer, ready for the outcome. It is similar to a perceptron in that it takes numerous inputs and creates a single output.

Considering DR imaging, several researchers applied a single ANN classification technique and got outstanding results. The authors of [63–68] employed a single ANN model and discovered that it was superior diabetes classifying strategy.

(h)    Unsupervised Classification: Unsupervised classification is employed when prior knowledge is unavailable. Inside this circumstance, just the set of information and characteristics that correspond to specific occurrences is revealed. In the chosen papers, unsupervised classification techniques were used many times. Zhou and Wu [22,69] used a ROC dataset with 100 images to perform unsupervised classification for microaneurysm identification. 50% of the photos were used for training, and the other 50% were used for testing in their experiments. In their experiments, the authors found that unsupervised classifiers performed reasonably well. Unsupervised classification

methods were used by Kusakunniran and Wu [70], and Biyani and Patre [53], to identify exudates in DR scans, with a sensitivity of 89% and 88%, accordingly.

(i)   Ensemble Classifiers: It is also called group learning and it combines different classification methods to create a more accurate model, and is a type of learning that takes place in groups [71]. There are three ways to do it: bagging, boosting, and stacking. Many classifying techniques work at a time in parallel during bagging, and the most accurate one is voted on at the end. The final classifier is the one that receives the most votes. Boosting is a technique that employs a sequence of classification algorithms. The weight of every model is adjusted based on the prior iteration. The data are split into many segments, each of which is checked with the help of others, and so on [72]. The stacking comprises base models, often known as level-0 models, as well as a meta-model that combines the level-0 model prediction. Stacking contrasts with boosting, in which a meta-model focuses on learning how to effectively combine the predictions again from base models, rather than a series of models that solve former models' prediction mistakes [73].

HDT with FFNN was used by Mane and Jadhav [74] to generate a categorization mechanism. The authors used two data sets, DIARETDB0 and DIARETDB1, to evaluate their DR image categorization capabilities over HDT and LMNN independently, but instead concluded that it was 98% reliable. Fraz and Jahangir [75] developed a classifier model using datasets DIARETDB1, e-Ophtha Ex, and Messidor. All 137 images were used for training their ML system and 341 images were used to evaluate their ensemble-based classifier. 98% accuracy was achieved in their experimentation.

(j)   Adaptive Boosting: AdaBoost is a systematic way to analyze a wide range of empirical systems. It works step-by-step, wherein each tree fits into a modified version of the original dataset before producing a robust classifier. This technique was utilized once in the chosen significant research. The AdaBoost method was used by Prentasic and Loncaric [76], wherein exudates were detected in DR images and a sensitivity of 75% was achieved, according to their experiments.

(k)   Self-Adaptive Resource Allocation Network Classifier: It selects training data based on self-regularized phenomena and then discards redundant data, requiring less memory and computer capacity. The network is then trained using the selected samples that have more information. Although the SRAN method was used two times in the primary ML tests, it did not perform as well as other categorization techniques. The authors of [22,77] evaluated the SRAN classification method to the McNN and SVM classifiers again to identify and track different ocular illnesses. A dataset from Coimbatore, India's Lotus Eye Hospital, was used in their study.

In ML-based techniques, better performance is given by statistically-based characteristics also followed by shape and structure [78]. ANN gives better performance for classification over SVM and, in the case of ML techniques, ensemble classifiers perform better. Deep learning is a part of machine learning which works with artificial neural networks. It requires a huge amount of data and gives more accurate results as compared to ML techniques. In DL, CNN is primarily applied to extract and categorize the DR images automatically. The further section discusses DL and DR in detail.

### 3.2. Deep Learning in DR Screening

DL is a subclass of ML that has grown towards a more robust and valuable tool, a practical methodology for ML. A DL model is comprised of a complex architectural style with a multidimensional framework. In medical image analyses, DL [79] is used to categorize, localize, segment, and identify medical images. With multiple methodologies, DL delivers more spectacular and promising DR disease diagnosis and categorization results. Convolutional neural networks (CNN), deep Boltzmann machines (DBM), autoencoders, deep neural networks (DNN), recurrent neural networks (RNN), deep belief networks (DBN), and generative adversarial networks (GAN) are only a few of the DL-based techniques [80].

CNN is more widely employed in medical imaging than other DL approaches. The three layers within CNN architecture are convolution, pooling, and fully connected layers. The CNNs dimensions, levels, and filter number could be changed to suit the author's requirements. Several filters combine in the convolutional layers to extract image features and build feature maps. Second, the average or max-pooling strategy is commonly used to reduce the volume of feature maps in pooling layers. Finally, the whole image feature set is created using completely connected layers. Subsequently, the data are categorized among two kernel functions, sigmoid (binary classification) or SoftMax (multi-classification) [80].

Whenever the sample data are insufficient, the data can be retrieved and precompiled to increase image features, and augmenting is performed as needed. The information is then examined. To sort Kaggle color images into five categories depending on DR scales, Pratt et al. [22] used a CNN containing ten convolutional layers, eight max-pooling layers, three fully connected layers, and a SoftMax classifier. Fundus images are shrunk and adjusted. L2 regularization and dropout methods proved helpful for the authors to minimize overfitting. They achieved a specificity of 95%, correctness of 75%, and sensitivity of 30%.

Wang et al. [81] combined handmade characteristics with CNN features. They then used the random forest (RF) classifier to develop a classifier that identifies exudate growths (hard) using the HEI-MED and E-ophtha datasets. Three convolutional and pooling layers were utilized to create the CNN, and only one FC layer was used for feature finding. HEI-MED and E-ophtha achieved an AUC of 0.9323 and 0.9644, respectively, and a sensitivity of 0.9477 and 0.8990. The DRIVE, STARE, and CHASE DB1 datasets were used to create a complete CNN model to capture blood vessels, as noted by Oliveira et al. [70]. The color fundus images were first refined. The dataset images were normalized using the stationary wavelet transform after extracting the green channel (SWT). Finally, before CNN processing, the patches were removed and augmented. In the DRIVE, STARE, and CHASE DB1 databases, the model achieved AUCs of 0.9821, 0.9905, and 0.9855, respectively. Chudzik et al. [82] used eighteen convolution layers, batch normalization layers, three max-pooling layers, up-sampling layers, and four skipped connections to construct a CNN model. Using image databases, the author employed three datasets to identify microaneurysms: E-ophtha, DIARETDB1, and ROC. Before CNN processing, images were preprocessed.

Yan et al. [83], introduced a technique for identifying DR lesions using DIARETDB1 by combining standard handcrafted features, improvised LeNet features, and a classifier and taking a combination of U-net and improved LeNet. During the preparatory step, green channels were clipped, CLAHE was used to increase contrast enhancement, a Gaussian filter was employed to reduce distortion, and morphological approaches were applied. The U-net design was utilized to segment the blood arteries when detecting red lesions. The LeNet architecture was upgraded with four convolutional layers, three max-pooling layers, and a fully connected layer to produce 48.71% sensitivity. DeepDR, a self-contained technique that combines pre-trained models Resnet and Inception V3, was proposed by Zhang et al. [39]. It has a sensitivity of 97.5% and a specificity of 97.3%, with an AUC of 97.7%.

DL models (DLMs) are a recent type of ML that uses a multi-layered artificial neural network (ANN) to learn more excellent representation from the information. DL has become the most common technique for detecting, predicting, forecasting, and classifying problems in various domains in recent years. It exposes numerous prospects for avoiding such a terrible condition in the healthcare profession, especially DR [12]. The DLMs were highly influential in various computer vision and biomedical image analysis applications [39,84]. It has proven to be an effective method of categorizing medical data. The CNN model plays a vital role in NPDR category classification, including excellent specificity and sensitivity using fundus photos. This also improved the effectiveness, availability, and affordability of DR grading systems, which were tested on several exciting images and scenarios [83,85]. Figure 10 shows the diagrammatic explanation of DR using DL.

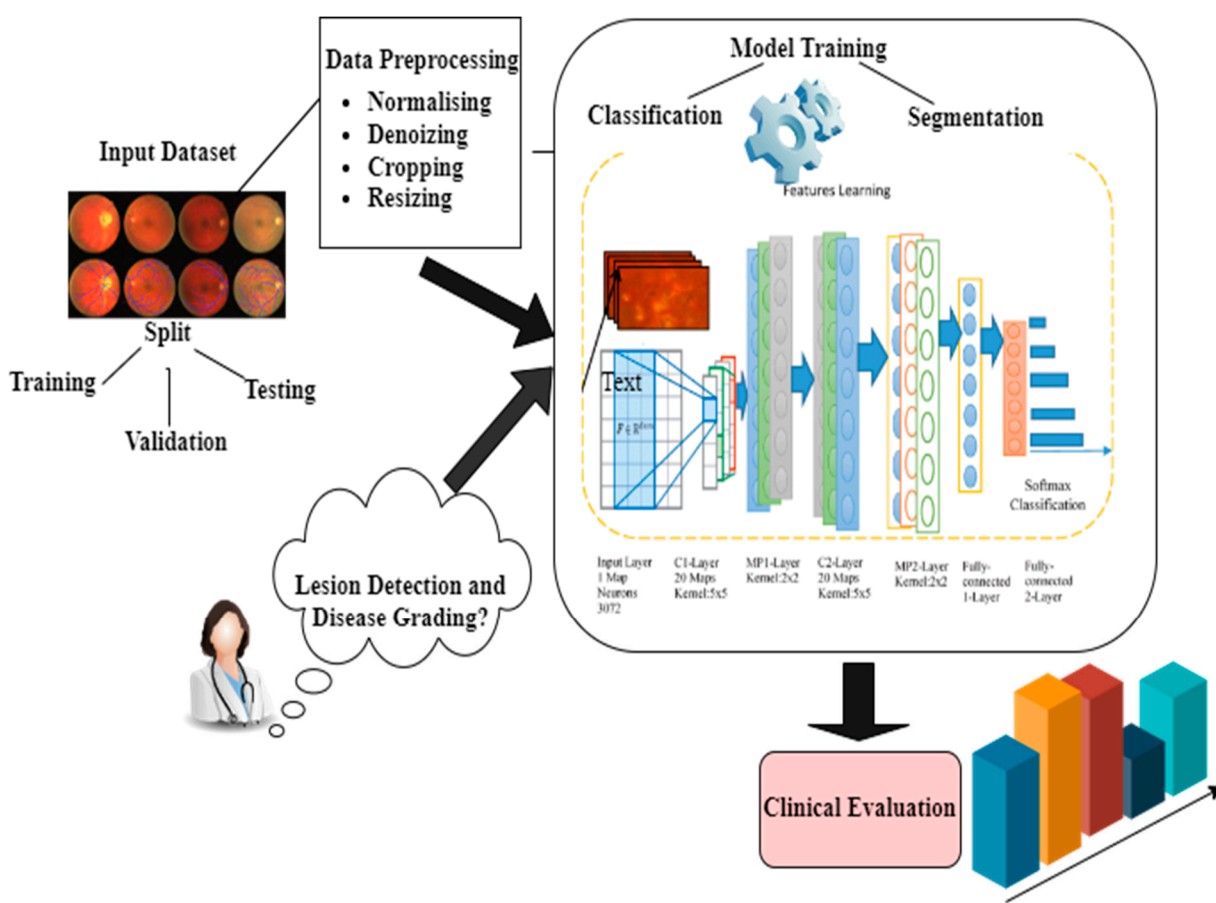

**Figure 10.** DR recognition and classification using deep learning.

Apart from the other DL approaches presented in the above section, transfer learning is a novel strategy that allows us to employ a previously trained model on a new problem statement. The topic of DR transfer learning is covered further down.

*3.3. Transfer Learning in DR*

Transfer Learning is a pretrained network, also known as a stored network, and is a typical DL approach that uses a limited set of feature examples. It improves the learning performance of a target task by transferring knowledge learned in a pre-trained network [86,87]. Transfer learning is utilized when inadequate data are required for the target task training and even when the perfect option for possible concerns is available. Applying this understanding to the target problem, transfer learning is employed to train a CNN by not re-initializing CNN weights [87]. Weights are instead loaded from a different CNN trained on a large dataset. The only healthy dataset to transfer trained version weights is the ImageNet problem [88].

There are two parts to pre-trained networks. The initial section consists of a succession of convolution and pooling layers followed by a densely linked classifier. Convolutional feature maps consider object locations in an input image. Object detection tasks, on either hand, are frequently worthless for densely coupled levels at the tip of the convolutional base. A pretrained network has been trained in a vast dataset, typically used for large-scale picture classification techniques. VGG19, Xception, MobilNet, ResNet, DenseNet, and Inceptionv3 are a few types of networks. The ImageNet dataset was used to train these networks, which consist primarily of everyday items and animals. By understanding the spatial ranking of characteristics, pre-trained systems might be used as generic models if the data are large and general enough [88]. A pretrained model could be used in feature extraction and fine-tuning. The convolutional base of a pretrained system with

new classifiers is used for feature extraction, and the dataset is processed via it. Since it learns general representations for various tasks, the convolutional foundation is reusable. Fine-tuning, performed in conjunction with feature extraction, entails unfreezing portions of the model's top layers (used for feature extraction). The top layers and the classifier of the system are then trained together [85,88]. Figure 11 depicts transfer learning from a pre-trained CNN model. The learned convolutional base can be used to extract features, which are then fed into a new classifier. Negative transfer and overfitting are significant difficulties when using full-scale, fine-tuned transfer learning systems. Whenever the domain of the pre-trained system and the performance are not equal, the reverse transfer happens, leading to low efficiency in the transfer learning model [89]. This can be due to the complexity of the characteristics collected in the last layers, which are unsuitable for the target domain. Fine-tuning subsequent layers, on the other hand, cause overfitting issues. To adjust later-level features to our goal domain, we must train the layer with such a huge set of parameters. It is not possible because the common, pretrained network InceptionV3 contains 21,802,784 parameters, while ResNet152 contains 58,370,944 parameters. Whenever these large-scale networks are trained, there is a risk of overfitting [87,88]. The success of using transfer learning for DR classifications may be analyzed by comparing trained models from inception to their fine-tuned variants.

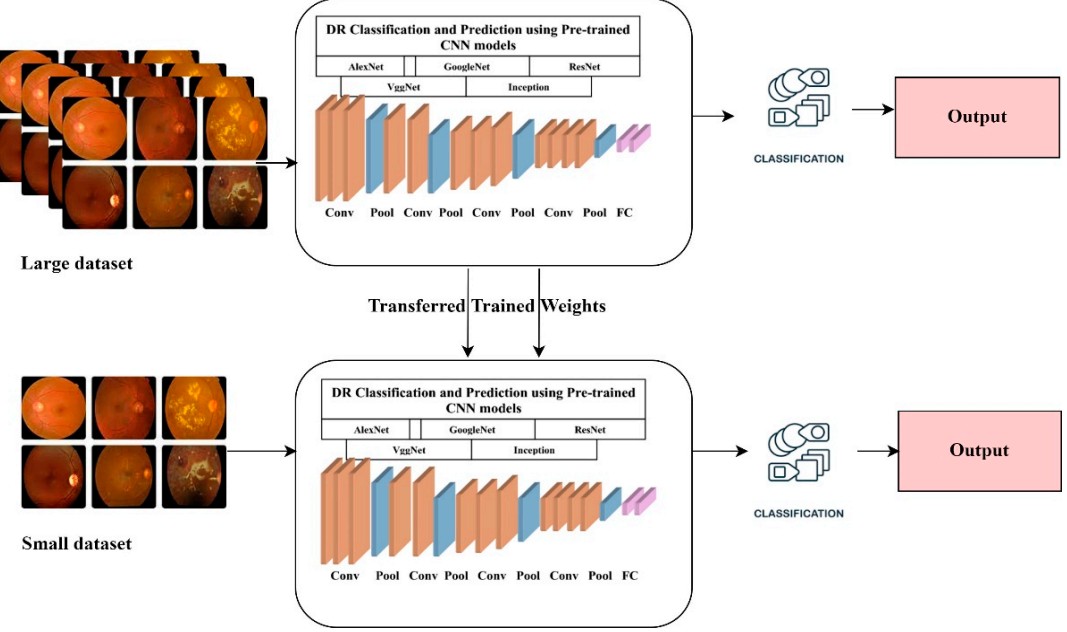

**Figure 11.** DR Detection and classification using transfer learning.

Several researchers in [90–92] concluded that utilizing transfer learning improves a model's accuracy in categorizing DR [93]. To train a DL model, many photos are required. The number of images in DR image datasets is restricted. Collecting DR photos and correctly labeling them is a time-consuming, experience-intensive, and resource-intensive operation [73].

## 4. RQ2 Feature Extraction Techniques for DR

### 4.1. Explicit or Traditional Feature Extraction Methods

The literature on image processing and image segmentation is researched a lot. These methods use shallow ML classifiers to diagnose DR using image processing-based feature detectors to measure blood vessels and the optic disc, and count abnormalities such as the presence of lesions, including hard exudates, soft exudates, microaneurysms, and hemorrhages, in an image. Additionally, features such as shape, color, intensity, statistical features, and texture-based features are examples of these characteristics. The following

paragraphs provide a summary of these characteristics and are also given in Figure 12 and Table 7 gives traditional feature extraction methods used in DR.

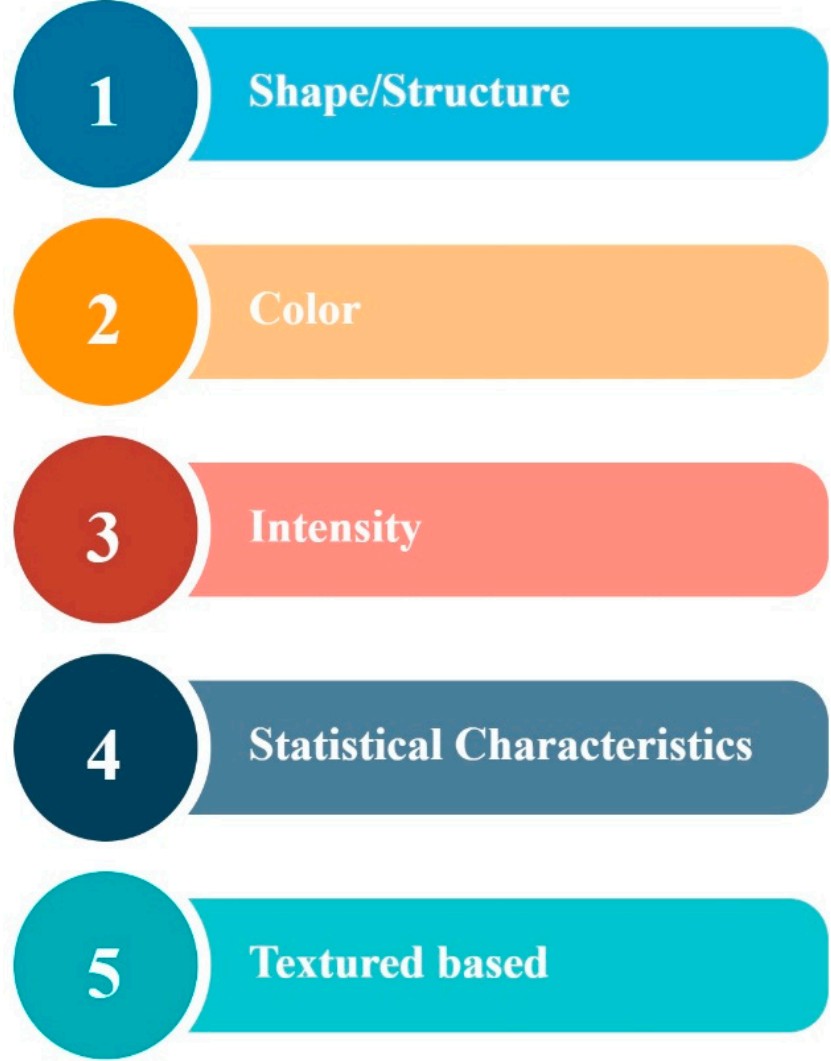

**Figure 12.** Feature extraction parameters.

(a) Characteristics based on shape and structure: The shape and size of various DR lesions, such as hard and soft exudates, hemorrhages, and microaneurysms, are among these characteristics. For example, Zhou and Wu [94] used shape-based criteria to detect microaneurysms: area and perimeter, axis length, circularity, and compactness.

(b) Features based on RGB (color): These characteristics are based on the image's red, green, and blue color planes. For example, authors Jaya and Dheeba [95] employed color fundus images to detect hard exudates and used four color-based characteristics. They used RGB color space to build color histograms.

(c) Features based on intensity: The pixel intensity in the red, green, and blue planes is called intensity. The authors of [11] employed intensity characteristics to detect cotton-wool patches in DR images. Similarly, the authors of [16] employed intensity characteristics to determine hard and soft exudates by computing maximum and lowest pixel intensities.

(d) Statistical characteristics: Statistical parameters are used to calculate statistical calculations from the pixels of a DR image. The authors of [96] used statistical and color features to identify hemorrhages in retinal images. The statistical parameters employed are maximum and minimum mean with standard deviation. Feature extraction plays a vital role in research methodologies. Feature extraction methods in the case of DR can

be classified as explicit or traditional feature extraction methods and direct methods or implicit methods (based on CNN).

(e) Texture-based characteristics: They provide texture-related information on DR images. Entropy, cluster shade, discrepancy, and correlation were four GLCM-based features that Vanithamani and Renee Christina identified [97]. Authors Nijalingappa and Sandeep [61] employed GLCM to extract textural features such as contrast, correlation, energy, homogeneity, similarity entropy, sum variance, difference variance, sum entropy, difference entropy, sum average, and inverse difference moment. In the selected papers, writers used several features in ML methodologies, such as form, shade, intensity, statistical, and texture-based characteristics. Shape and statistical features are the most frequently employed combination of features.

**Table 7.** Traditional feature extraction methods used in DR.

| Ref. No | Authors | Feature Selected | Features and Classifiers (Technique) | Weakness | Database | (Performance Analysis) |
|---|---|---|---|---|---|---|
| [98] | Di Wu, Wu, Zhang, Liu, and Bauman, 2006. | To find out blood vessels in the retina. | Gabor filters. | Requires high-performance time with greater feature vector dimension. | STARE. | Tested 20 images. For normal images, TPR—80 to 91% and FPR—2.8 to 5.5%. For abnormal images, TPR—73.8–86.5% FPR—2.1–5.3%. |
| [99] | Sanchez et al. (2009). | To detect hard exudates from cotton wool spots and other artifacts. | Edge detection and mixture models. | The diversity of brightness and size makes it difficult to detect the hard exudates, hence method may fail when they appear very few in the retina. | Eighty retinal images with variable color, brightness, and quality. | A sensitivity of 90.2% and a positive predictive value of 96.8% for an image-based classification accuracy sensitivity of 100% and a specificity of 90%. |
| [100] | Garcia, Sanchez, Lopez, Abasolo, and Hornero (2009). | Red lesions image and shape features. | Neural networks with multilayer perceptron (MLP), radial basis function (RBF), and support vector machine (SVM). | The black box nature of ANN and more accuracy requires more amount of data. | The database was composed of 117 images with variable color, brightness, and quality. 50 were used for training and 67 for testing. | Using lesion-based sensitivity and positive prediction values in percent. MLP—88.1, 80.722. RBF—88.49, 77.41. SVM—87.61, 83.51. Using image-based sensitivity and specificity in percent. MLP—100, 92.59. RBF—100, 81.48. SVM—100, 77.78. |
| [101] | Sanchez et al. (2008). | Hard exudates. | Color information and Fisher's linear discriminant analysis. | When there are only a few very faint HEs in the retina, this proposed algorithm may have limited performance. More images are required for better results. | Fifty-eight retinal images with variable color, brightness, and quality from the Instituto de Oftalmobiología Aplicadaat University of Valladolid, Spain. | Using a lesion-based performance sensitivity of 88% with a mean number of $4.83 \pm 4.64$ false positives per image. Using Image-based sensitivity-100 and Specificity of 100% is achieved. |
| [102] | Quellec et al. (2012). | Abnormal patterns in fundus images. | Multiple-instance learning. | The training procedure is complex and takes a lot of time. | Messidor (1200 images) and e-optha (25,000 images). | In the Messidor dataset, the proposed framework achieved an area under the ROC curve of $A_z = 0.881$ and e-optha $A_z = 0.761$. |

**Table 7.** *Cont.*

| Ref. No | Authors | Feature Selected | Features and Classifiers (Technique) | Weakness | Database | (Performance Analysis) |
|---|---|---|---|---|---|---|
| [103] | Kose, ˌSEvik, ˙IKibaˌs, and Erdo¨l (2012). | Image pixel information. | Inverse segmentation using region growing, adaptive region growing, and Bayesian approaches. | Difficult to choose the correct way to select a prior. | A total of 328 images with 760 X 570 resolution from the Department of Ophthalmology at the Faculty of Medicine at Karadeniz Technical University were used. | This approach successfully identifies and localizes over 97% of ODs and segments around 95% of DR lesions. |
| [104] | Giancardo et al. (2012). | Exudates in fundus images. | Feature cector generated using an exudate probability map, the color analysis, and the wavelet analysis. Exudate probability map and wavelet analysis. | Intensive calculation. | HEI-MED, Messidor, and DIARETDB1. | AUC is between 0.88 and 0.94, depending on the dataset/features used. |
| [105] | Zhang, Karray, Li, and Zhang (2012). | Microaneurysms and blood vessel detection. | Locate MAs using multi-scale Gaussian correlation filtering (MSC) with dictionary learning and sparse representation classifier (SRC). | Dictionaries for vessel extraction are artificially generated using Gaussian functions which can cause a low discriminative ability for SRC. Additionally, a larger dataset is required. | STARE and DRIVE. | For STARE: FPR—0.00480. TPR—0.73910. PPV—0.740888. For DRIVE: FPR—0.0028. TPR—0.5766. PPV—0.8467. |
| [106] | Qureshi et al. (2012). | Identifying macula and optic disk (OD). | Ensemble combined algorithm of edge detectors, Hough transform, and pyramidal decomposition. | It is difficult to determine which one is the best approach because good results were reported for healthy retinas but less precise on a difficult data set. | Diaretdb0, Diaretdb1, and DRIVE 40% of the images from each benchmark are used for training and 60% of the images are used for testing. | The average detection rate of macula is 96.7 and OD is 98.6. |
| [107] | Noronha and Nayak (2013). | Two energy features and six energy values in three orientations. | Wavelet transforms and support vector machine (SVM) kernels. | The performance depends on factors such size and quality of the training features, the robustness of the training, and the features extracted. | Fundus images were used. | Accuracy, sensitivity, and specificity of more than 99% are achieved. |
| [108] | Gharaibeh N (2021). | Cotton wool spots, exudates. Nineteen features were extracted from the fundus image. | Unsupervised particle swarm optimization based relative reduct algo (US-PSO-RR), SVM, and naïve-Bayes classifiers. | Detection and elimination of optic discs from fundus images are difficult, hence lesion detection is challenging. | Image-Ret. | Obtained a sensitivity of 99%, A specificity of 99% and a high accuracy of 98.60%. |

**Table 7.** *Cont.*

| Ref. No | Authors | Feature Selected | Features and Classifiers (Technique) | Weakness | Database | (Performance Analysis) |
|---|---|---|---|---|---|---|
| [109] | Gharaibeh N (2018). | Microaneurysm, hemorrhage, and exudates. | Co-occurrence matrix and SVM. | Can be tried on larger datasets. | DIARETDB1. | Obtained a sensitivity of 99%, a specificity of 96%, and an accuracy of 98.4%. |
| [110] | Akram, Khalid, and Khan (2013). | Image shape and statistics. | Gaussian mixture models and support vector machine and Gabor filter bank. | Need to work on a large dataset. | Four hundred and thirty-eight Fundus images. | An accuracy of 99.4%, a sensitivity of 98.64%, and a specificity of 99.40% are achieved. |
| [111] | Harini R and Sheela N (2016). | Blood vessels, microaneurysms, and exudates. | The gray level co-occurrence matrix (GLCM) is utilized to extract textural features the classification is completed using SVM. | Problem working with large datasets since training requires more time with SVMs. | Seventy-five Fundus images were considered, forty-five were used for training, and thirty for testing | An accuracy of 96.67%, a sensitivity of fundus of 100%, and a specificity of 95.83% are achieved. |
| [112] | Anjana Umapathy, Anusha Sreenivasan, Divya S. Nairy (2019). | Exudates and red lesions in the fundus image. | Decision tree classifier. | Requires more time for training and persistent overfitting. | STARE, HRF, MESSIDOR, and a novel dataset created from Retina Institute of Karnataka. | The approach achieved an accuracy of 94.4%. |

### 4.2. Direct Methods

Direct methods are also called implicit methods. In recent works, direct methods are likely to use deep CNN. These direct methods do not need to extract features manually; instead, they acquire the patterns of DR anomalies and deliver grading results according to the grading criteria. CNN architecture such as ImageNet AlexNet, GoogleNet, and Inception-V3 are all used to train DR pictures in the literature. Furthermore, we do not need to create a feature vector with direct methods. These techniques are new research in the literature.

Zago et al. [113] used two CNNs to build an approach for detecting DR vs. non-DR color images based on the expectation of lesion regions (pre-trained VGG16 and CNN). A DIARETDB1 dataset was utilized for training. The datasets IDRiD, Messidor, Messidor-2, DDR, DIARETDB0, and Kaggle were used for checking. The Messidor data yielded the highest scores, with an AUC of 0.912 and a sensitivity of 0.94. Jiang et al. [114] generated a method that classified the fundus image dataset as referable DR or non-referable DR using three CNNs (Inception-v3, ResNet152, and Inception-ResNet-v2). The photos were scaled, improved, and augmented before CNN training. The Adaboost approach would then be used to connect them. All network weights were modified using the Adam optimizer, and the model had a correctness of 88.21% and an AUC of 0.946, to estimate the five-stage DR and evaluate the performance of CNNs.

Wang et al. [75] used the Kaggle fundus dataset and three distinct CNNs, namely (pre-trained VGG16, AlexNet, and Inception-v3). With all three pre-trained models, the fundus images have been resized to different forms, yielding 63.23%, 50.03%, and 37.43% using Inception-v3, VGG16, and AlexNet, accordingly.

Hua et al. [111] used the DRIVE dataset photos to extract the retinal blood vessels. The author chose four feature maps using a ResNet-101 pre-trained network; individual feature maps were then blended to create a single map. Before CNN processing, the fundus images were enhanced. With the best feature maps merged, the accuracy was 0.951, the sensitivity was 0.793, the AUC was 0.9732, and the specificity was 0.9741. The retinal blood vessels were extracted using a CNN created by Wu et al. [112] from three well-known databases: STARE, DRIVE, and CHASE. In the preprocessing phase, the color images were

transformed to grayscale images, normalized, augmented, and the image contrast was increased using CLAHE. For datasets such as STARE, DRIVE, and CHASE, the model attained AUCs of 98.75%, 98.30%, and 98.94%, respectively, and an accuracy of 96.72%, 95.82%, and 96.88%.

## 5. RQ3 Datasets Available for DR

An enormous number of public datasets are available for DR. Training, validation, and testing can be performed using various datasets, and the performance of various systems can be compared and examined. Fundus color photographs and optical coherence tomography (OCT) are retinal imaging methods. OCT gives the internal structure of the retina and is available in 2D and 3D. In contrast, images taken from fundus cameras are 2D photographs of the retina [115]. The Table 8 gives various fundus image datasets:

**Table 8.** Datasets.

| Sr. No | Dataset Name | Description | References | Availability | Link |
|---|---|---|---|---|---|
| 1 | Kaggle | EyePACS has supplied this dataset for the DR detection challenge. There are 88,702 photos in this collection (35,126 for training and 53,576 for testing) [116]. | [31,49,55,56,116–125] | Free | https://www.kaggle.com/c/diabetic-retinopathy-detection/data (accessed on 2 May 2022). |
| 2 | ROC (Retinopathy Online Challenge) | There are 100 photos in this collection. Canon CR5-45NM, Topcon NW 100, and NW 200 cameras were used. | [52,57,60,69,82,126,127] | Free | http://webeye.ophth.uiowa.edu/ROC/ (accessed on 2 May 2022) |
| 3 | DRIVE | This dataset contains 40 photos from a DR program in Holland (split into training and testing, 20 images each). The camera was a Canon CR5 non-mydriatic 3CCD with a 45-degree field of view (FOV). | [57,65,128–133] | Free | https://www.isi.uu.nl/Research/Databases/DRIVE/Gulshan (accessed on 2 May 2022) |
| 4 | STARE | There are 400 photos in total in this dataset. The fundus camera used was a Topcon TRV-50 with a 35-degree field of view. | [57,128,130,132–136] | Free | http://www.cecas.clemson.edu/~ahoover/stare/ (accessed on 3 May 2022) |
| 5 | E-Optha | The OPHDIAT telemedical network created this dataset. E-Ophta MA and E-Ophta EX are the two datasets that make up this collection. Both have 381 and 82 photos in them, respectively. | [55,70,75,82,96,137–139] | Free | http://www.adcis.net/en/Download-Third-Party/E-Ophtha.html (accessed on 3 May 2022) |
| 6 | DIARETDB0 | There are 130 photos in this dataset (normal images = 20, images with DR symptoms = 110). The photos were obtained with a fundus camera with a field of view of 50 degrees. | [55,61,74,140] | Free | http://www.it.lut.fi/project/imageret/diaretdb0/ (accessed on 3 May 2022) |
| 7 | DIARETDB1 | There are 89 photos in this dataset (standard images = 5, images with at least mild DR = 84). The photos were obtained with a fundus camera with a field of view of 50 degrees. | [53,55,57–60,62–64,67,70,74,75,82,96, 97,119,135,137,139,141–145] | Free | http://www.it.lut.fi/project/imageret/diaretdb1/index.html (accessed on 4 May 2022) |
| 8 | Messidor-2 | This dataset includes 1748 photos collected with a Topcon TRC NW6 non-mydriatic fundus camera with a 45-degree field of view. | [146] | On-demand | http://www.latim.univ-brest.fr/indexfce0.html (accessed on 3 May 2022) |
| 9 | Messidor | This dataset includes 1748 photos collected with a Topcon TRC NW6 non-mydriatic fundus camera with a 45-degree field of view. | [58,61,66,75,97,125,137,141,147–150] | Free | http://www.adcis.net/en/Download-Third-Party/Messidor.html (accessed on 3 May 2022) |
| 10 | DRiDB | This dataset, which includes 50 photos, is accessible upon request. | [76,94] | On-demand | https://www.ipg.fer.hr/ipg/resources/image_database (accessed on 3 May 2022) |
| 11 | DR1 | The Department of Ophthalmology of the Federal University of Sao Paulo created this dataset. (UNIFESP). It contains 234 images captured with TRX-50X, the mydriatic camera having 45 degrees FOV. | [54,150] | Free | http://www.recod.ic.unicamp.br/site/asdr (accessed on 4 May 2022) |
| 12 | DR2 | The Department of Ophthalmology at the Federal University of Sao Paulo also contributed to this dataset (UNIFESP). It contains 520 photographs taken with the TRC-NW8, a non-mydriatic camera with a 45-degree field of view. | [54] | Free | http://www.recod.ic.unicamp.br/site/asdr (accessed on 3 May 2022) |
| 13 | ARIA | This dataset contains 143 images. The camera used was a Zeiss FF450+ fundus camera with a 50-degree field of view. | [151] | Free | http://www.damianjjfarnell.com/?page_id=276 (accessed on 5 May 2022) |
| 14 | FAZ (Foveal Avascular Zone) | There are 60 photos in this dataset (25 images that are normal and 35 images with DR). | [141] | Free | http://www.biosigdata.com/?download=Zone (accessed on 5 May 2022) |
| 15 | CHASE-DB1 | There are 28 photos of 14 children included in this dataset (consisting of one image/eye). CHASE-DB1 deals with Child Heart and Health Study (CHASE) in England. | [130] | Free | https://www.blogs.kingston.ac.uk/retinal/chasedb1/ (accessed on 5 May 2022) |
| 16 | Tianjin Medical University Metabolic Diseases Hospital | This dataset contains 414 fundus images. | [57] | Not publicly available | http://eng.tmu.edu.cn/ResearchCenter/list.htm (accessed on 5 May 2022) |
| 17 | Moorfields Eye Hospital | Data from countries such as Kenya, Botswana, Mongolia, China, Saudi Arabia, Italy, Lithuania, and Norway are collected at Moorfields Eye Hospital in London. | [60] | Not publicly available | https://www.moorfields.nhs.uk/research-and-development (accessed on 5 May 2022) |

**Table 8.** *Cont.*

| Sr. No | Dataset Name | Description | References | Availability | Link |
|---|---|---|---|---|---|
| 18 | CLEOPATRA | The CLEOPATRA collection consists of 298 fundus images. It includes images from 15 hospitals across the United Kingdom to diagnose DR. | [152] | Not publicly available | Not available |
| 19 | Jichi Medical University | There are 9939 posterior pole fundus images of diabetic patients in this dataset. The camera used was a NIDEK Co., Ltd., Aichi, Japan, AFC-230, with a 45-degree field of view. | [153] | Not publicly available | https://www.jichi.ac.jp/ (accessed on 5 May 2022) |
| 20 | Singapore National DR Screening Program | This dataset was collected during the Singapore National Diabetic Screening Program (SIDRP) between 2010 and 2013; a total of 197,085 retinal images were collected. | [97] | Not publicly available | Not available |
| 21 | Lotus Eye Care Hospital Coimbatore, India | It contains 122 fundus images (normal = 28, DR = 94). A Canon non-mydriatic Zeiss fundus camera with a FOV of 90 degrees was used. | [22,77,154] | Not publicly available | https://www.lotuseye.org/centers/sitra/ (accessed on 5 May 2022) |
| 22 | Department of Ophthalmology, Kasturba Medical College, Manipal, India | This dataset contains 340 images (normal = 170, with retinopathy = 170). Non-mydriatic retinal camera, namely, TOPCON, was used | [155] | Not publicly available | https://manipal.edu/kmc-manipal/department-faculty/department-list/ophthalmology.html (accessed on 5 May 2022) |
| 23 | HUPM, Cádiz, Spain | Fundus photos from Hospital Puerta del Mar in Spain were taken, including 250 photos (50 normal and 200 with DR symptoms). | [156] | Not publicly available | https://hospitalpuertadelmar.com/ (accessed on 5 May 2022) |

## 6. RQ4 Evaluation Measures Used for DR Detection

Performance measure parameters play a vital role in evaluating a model on futuristic or unknown data to estimate a model's generalization accuracy. A few of them are listed below.

### 6.1. False Positive Rate (FPR)

The percentage of times that segmenting retinal images produces positive instead of negative findings.

It can be written as follows:

$$\text{FPR} = \frac{\text{FP}}{\text{TN} + \text{FP}} \tag{1}$$

### 6.2. False Negative Rate (FNR)

It is the percentage of time that segmenting retinal images produces negative instead of positive findings.

It can be written as follows:

$$\text{FNR} = \text{FN}/(\text{TP} + \text{FN}) \tag{2}$$

### 6.3. Accuracy [89]

It is the ratio of correctly assigned pixels in the segmented image to several blood vessel pixels.

It is given as:

$$\text{A} = \frac{\text{TN} + \text{TP}}{\text{TN} + \text{FN} + \text{FP} + \text{TP}} \tag{3}$$

### 6.4. Specificity

It is the ratio of correctly detected vessels to the total number of non-vessels.

$$\text{Spec} = \frac{\text{TN}}{\text{FP} + \text{TN}} \tag{4}$$

### 6.5. Sensitivity/Recall Rate

It is the ratio of accurately identified vessels to the total number of vessels.

$$\text{Sen} = \frac{\text{TP}}{\text{FN} + \text{TP}} \tag{5}$$

### 6.6. F-Score

The F-score is a measurement of test accuracy. The number of positive outcomes is divided by the number of authentic positive results.

$$F - Score = 2 \times \frac{Recall \times Precision}{Recall + Precision} \tag{6}$$

### 6.7. ROC

It is a graph showing classifier performance at all conceivable thresholds. The graph depicts the positive rate (on the *Y*-axis) and the false positive rate (on the *X*-axis).

### 6.8. Positive Predictive Value (PPV)

It can be given by the probability of fundus images being segmented accurately.

### 6.9. Negative Predictive Value (NPV)

It can be given by the probability of fundus images being segmented inaccurately.

### 6.10. False Discovery Rate (FDR)

It's also known as a false positive and can be defined as the rate of an expected part of errors.

### 6.11. Confusion Matrix

A confusion matrix is used to find out what our ML algorithm achieved and where it went wrong. It is a matrix used for evaluating classification models' performance on a given set of test data. It can only be determined if the values for testing data are initially known. It is also known as an error matrix since it displays the flaws in the model's performance as a matrix [157].

The following are some characteristics:

(a) Rows correspond to what is predicted and columns correspond to the known truth or actual values. Here, a matrix for the prediction of two classes for a classifier is given by a 2 × 2 table, three classes by a 3 × 3 table, etc.

(b) Actual values are the actual values for the given observations, whereas projected values are predicted by the model.

(c) The following Table 9 gives the values,

**Table 9.** Sets ed.

| N = Total Predictions | Actual: NO | Actual: Yes |
|:---:|:---:|:---:|
| Predicted: No | True Negative | False Positive |
| Predicted: Yes | False Negative | True Positive |

The following cases are listed in the table above.

1. True Negative: when the model's predicted and the actual value is No.
2. True Positive: when the model's predicted and the actual value is Yes.
3. False Negative: when the model's predicted value is Yes, and the actual value is No. It is also known as a Type-II mistake.
4. False Positive: when the model's predicted value is No, and the actual value is Yes. A type-I mistake is another name for it.

### 6.12. Kappa Value

Cohen's Kappa is a common statistic for determining how well two raters agree. It can also be used to evaluate a classification model's performance. Using Kappa, a comparison of ML model predictions to the humanly established credit scores can be made. Similar to

many other evaluation measures, Cohen's Kappa is calculated using the confusion matrix. On the other hand, Cohen's Kappa considers imbalances in class distribution and might be more challenging to assess than overall accuracy [158].

## 7. EMR and Biomarkers in DR

Biomarkers are biological elements found in blood, other bodily fluids, or tissues that indicate the presence of a good or aberrant process, as well as a condition or disease. They act as a highlighter and can be used to assess how well the body reacts to treatment for a symptom. They play an essential role in identifying the physiological state or disease detection. Measurable indicators such as blood pressure, temperature, C-reactive protein, etc., are examples of a few biomarkers. Circulating biomarkers may be beneficial in diagnosing initial retinal illness before structural reforms are visible with existing imaging technology [159]. Personalized diabetes vision care precisely forecasts the threat of diabetic retinopathy (DR) development and loss of vision in real-time [160]. This utilization of electronic medical records (EMR) provides a framework for the incorporation of artificial intelligence (AI) algorithms that anticipate DR development into healthcare decisions [161]. The threat of retinopathy evolution and vision problems can be projected using an algorithm applied to pieces of information from each patient, enabling patients to obtain prompt therapy. Hemoglobin A1c, also called HbA1c levels, are among the most well-known indicators for glycemic control. Hemoglobin A1c has been demonstrated to have a high correlation with the evolution of systemic symptoms of diabetes, particularly DR [162]. Early treatment diabetic retinopathy study (ETDRS) and the diabetic retinopathy severity scale (DRSS) are traditional biomarkers for DR. Longitudinal clinical studies show that DRSS-based fundus photographic assessment effectively represents the projected risk of disease development, responsiveness to treatments, and long-term visual results [3]. The DRSS has affected DR patients globally, employing a scale ranging from no diabetic retinopathy (NPDR) to severe proliferative diabetic retinopathy (PDR). Different ocular biomarkers consist of various parameters found in ocular coherence tomography (OCT), retinal blood flow, retinal oxygen saturation, vascular endothelial growth factor (VEGF), neural retina assessments (electroretinograms), and retinal vessel geometry [161,163]. Some novel biomarkers include:

- Genetics: The investigation of genes associated with the development of advanced DR, vascular endothelial growth factor (VEGF), lipoproteins, and inflammation. There have been genome-wide association studies and single nucleotide polymorphisms (SNPs) linked to an enhanced danger of sight-threatening retinopathy [164].
- Epigenetics: It is the study of how environmental variables interact with genes. DNA methylation, histone modification, and microRNAs are among the biomarkers being investigated [165,166].
- Proteomics: It is the study of protein structure and function research in cultured cells and tissues. A current study shows that diabetic patients have higher levels of transport proteins (vitamin D binding protein), arginine N-methyltransferase 5, and inflammatory proteins (leucine-rich alpha-2-glycoprotein) [167,168].
- Metabolomics: The study of chemical traces left by biological activities. Data on increased metabolite cytidine, cytosine, and thymidine found in DR patients using mass spectrometry is included in the studies. These nucleotide concentrations may be relevant in monitoring DR progression and evaluating therapy [169].

## 8. RQ5 Challenges and Future Research Directions

This section addresses several scientific issues that previous diabetic detection investigations have not addressed. Much effort is required to improve the effectiveness of various diabetic detection systems. Various research challenges and their workable solutions are listed below.

Challenge 1: The origins of DL models are frequently unknown; hence, they are viewed as a black box. This results in a need for an automatic (parameter) optimization

strategy. It is also challenging to determine the best configuration and values for layer numbers and node numbers in different layers. Basic domain knowledge is also required for selecting parameters for the number of epochs, learning rate, and to regularize strength. As a result, automatic optimization methodologies for various DL architecture parts for specific DM datasets and additional clinical datasets may be introduced.

Future Research Direction: Explainable artificial intelligence (XAI).

Recent developments in DL techniques have aroused interest in using AI technology in every field; however, the method's opacity has raised concerns regarding its use in security applications. The "explainability" component is critical since it demonstrates how black-box methods operate and provides accountability and accessibility aspects that regulations, consumers, and network operators care about. Explainable artificial intelligence (XAI) is indeed a collection of technologies and methodologies for transforming so-called black-box AI algorithms into white-box algorithms, wherein the outcomes of the methodologies and the variables, parameters, and measures adopted by the algorithm show up in such results have been transparent and straightforward [170]. There are three dimensions to evaluate when analyzing the comprehensiveness of AI models, as stated below.

Explainability is a learning model feature that allows the model's processes to be explained in detail. The strategy is to make the learning model's internal workings increasingly transparent. It is worth mentioning that sensitive applications necessitate explanation ability for both scientific interest's purpose and since the danger component takes priority over other factors whenever human lives are threatened. A learning model's interpretability is a factor that helps people to understand and make logical sense of it, as opposed to explainability. Transparency is typically associated with understandability; a learning model is said to be transparent in it can be understood without the need for an interface. The term "transparent" is defined as a communication paradigm that may be comprehended without additional elements.

For the objective of categorizing DR severity using color fundus photography, G. Quellec [151] describes explanatory artificial intelligence (XAI), which achieves the same level of performance as black-box AI (CFP). The algorithm learns to segment and categorize lesions in images, and the final image-level classification is derived directly from these multivariate lesion segmentations. The peculiarity of this explanatory framework is that similar to black-box AI algorithms, it is trained from the beginning to the end with only image supervision; the notions of lesions and lesion categories develop independently. Se-In Jang1 [171] describes a classification model of a neural-symbolic learning-based explainable DR (ExplainDR). To accomplish explainability, the authors develop a human-readable symbolic representation that follows a taxonomy style of DR characteristics connected to eye health issues. The disease prediction then incorporates human-readable information gained from the symbolic representation. There are various XAI models such as LIME, the What-If-Tool, DeepLIFT, Skater, SHAP, AIX360, Activation Atlases, Rulex Explainable AI, and GradCAM [172]. The XAI model's shapley additive explanations (SHAP) with guided backpropagation (GBP), and the inception-V3 framework have been used for ophthalmic diagnosis [173]. Diagramatic explanation of DR using XAI is shown in Figure 13.

Challenge 2: Insufficient and unlabeled data availability.

Future Research Direction: For training, DL algorithms often require a large amount of labeled diabetes data, hence there is a need to develop labeled and sufficient datasets for training. When the training range is restricted, it is impossible to achieve sufficient precision. This problem can be approached in two ways. To begin, low-learning algorithms were used to collect training data. Second, various enhancement techniques were used, including cropping, rotating, flipping, and color casting. More research is needed to generate more precise training data so that the DL design with more consistency and distinguishing features.

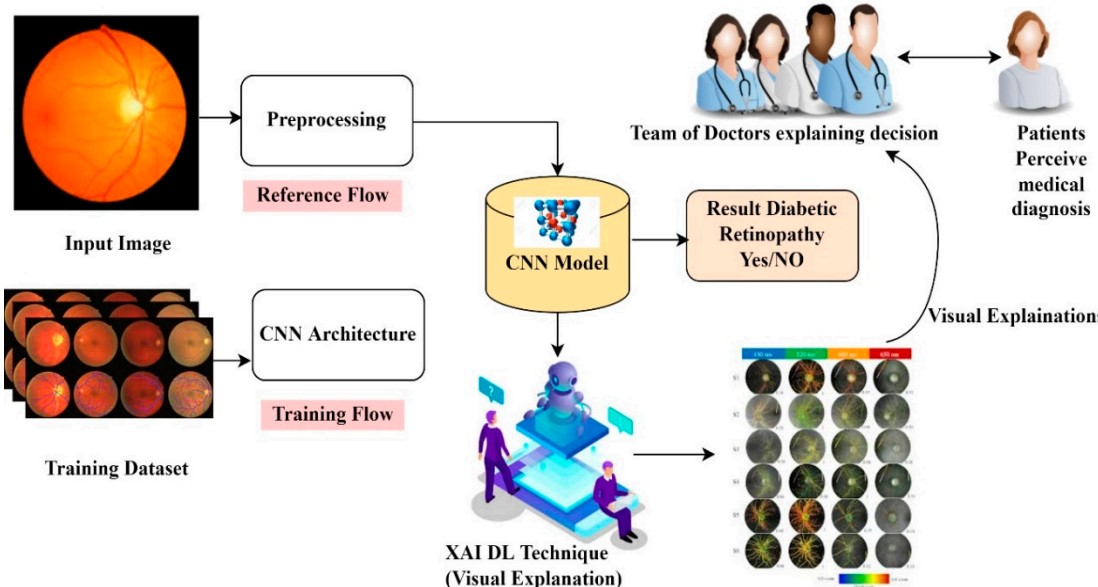

**Figure 13.** Diabetic retinopathy using XAI.

Challenge 3: Practitioners, especially ophthalmologists, prefer making decisions by referring to OCT, fundus, and other modalities. Hence, using multimodality in DR will help in detecting the severity of DR.

Future Research Direction: Recently, images such as fundus are used to diagnose eye-related disorders, such as DR and glaucoma, and OCT is used to detect other eye-related disorders such as diabetic macular edema and age-related macular degeneration, etc. [174]. There are many scopes to develop a similar architecture that is extensible enough to accommodate fundus images and OCTs for DR identification. Most available investigations used a single modality to build a DR detection model. In the future, however, a multi-modal concept can be used to view DR detection from more than one data perspective. This will boost the practitioner's confidence in detecting DR early.

Challenge 4: Lack of self-supervised or unsupervised approaches.

Future Research Direction: Domain adaptation applies a technique learned inside one domain to a new target domain. Duy M. H. Nguyen [175] tackles the topic of domain adaptation for DR grading by creating a novel self-supervised task based on retinal vascular image reconstructions inspired by medical domain knowledge to learn invariant target-domain features. Then, a benchmark of current state-of-the-art unsupervised domain adaptation approaches is offered on the DR problem. It has been demonstrated that their method outperforms other domain adaptation methodologies. In [176], Ruoxian Song proposed a domain adaption for multi-instance learning for DR grade, which organized weakly supervised DR grade as a multi-instance learning issue. Cross-domain produces tagged examples to filter out irrelevant examples in the target domain. To model the link between suspicious occurrences and bag labels, multi-instance learning with only an attention mechanism can collect location information of highly suspected lesions and predict the grade for DR. The proposed technique is tested on the Messidor dataset. The results showed an average accuracy of 0.764 and an AUC value of 0.749. Domain adaptation in DR is diagrammatically shown in Figure 14.

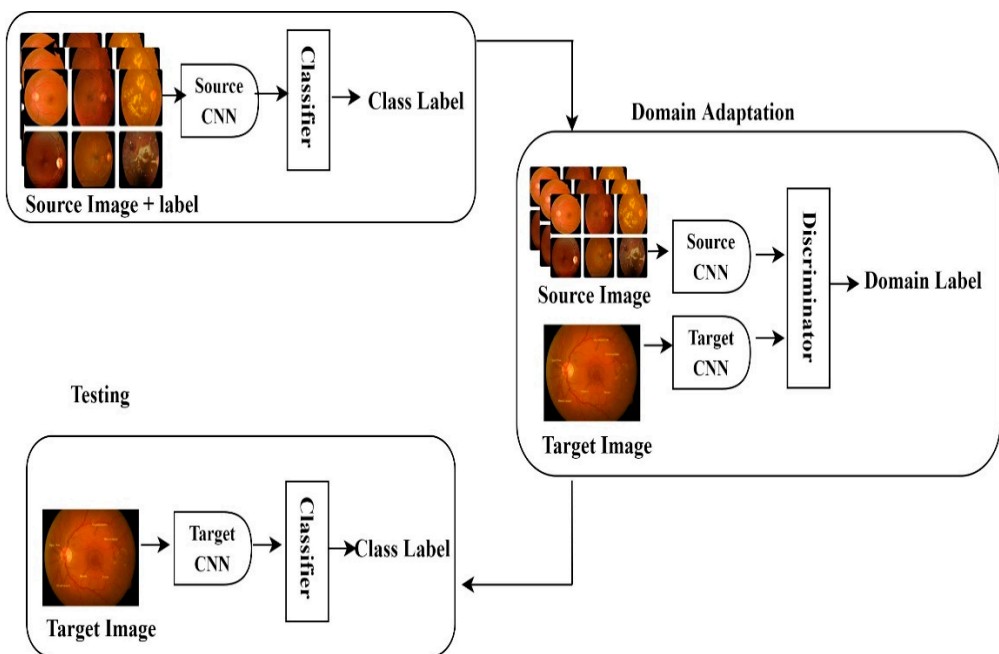

**Figure 14.** Diabetic retinopathy using domain adaptation.

Challenge 5: The need for improved data efficiency, less overfitting through common representations, and fast learning using auxiliary information in DR is required.

Future Research Direction: Multi-task learning (MTL) is a subfield of ML in which a shared model learns many tasks simultaneously. Improved data efficiency, less overfitting through common representations, and fast learning using auxiliary information are advantages of such techniques. On the other hand, simultaneous learning of many tasks poses modern design and optimization issues, and deciding which tasks should be learned together is a non-trivial problem in and of itself [177]. Although DL for DR screening has demonstrated incredible healthcare accuracy of referable versus non-referable DR, extra fine-grained grading of the DR severity level and automated segmentation of lesions (if any) in retina images will still be necessary. To conduct the DR grading and lesion segmentation tasks, A. Foo and W. Hsu [178] used a multi-task learning strategy using MTUnet. They proposed a semi-supervised learning technique to acquire segmentation masks for enormous datasets due to the lack of ground truths for lesion segmentation masks. A. Foo and W. Hsu [178] discovered that the DR severity level of an image could be influenced by the existence and prevalence of several lesions. Experimentation was performed on publicly accessible datasets and records, and data produced via screening programs indicate the efficacy of the multi-task approach over the state-of-the-art network.

Challenge 6: There seems to be currently no unified standard for assessing and validating AI algorithms. The testing sets used in numerous studies vary substantially. A few other studies did not use independent external testing sets but instead used internal validation sets to test the algorithm's sensitivity, specificity, and AUC. Various studies' sensitivity, specificity, AUC, and other indicators just were not comparable. As a result, standard testing is necessary to analyze each algorithm.

Future Research Direction: The need for a standard testing technique to analyze algorithms.

Challenge 7: In the most recent findings, one AI system can only identify one disease, implying that a patient can only be evaluated for a single issue during a fundus examination. However, if we consider the fundus image modality, almost all retinal vascular diseases can be examined through it, unless the media is hazy, so the image is not clear [35]. The eye examination process will be greatly simplified if an AI system can diagnose multiple diseases. Detection of other eye diseases during DR screening has been reported

in studies, which can detect age-related macular degeneration, as well as other diseases, at the same time.

Future Research Direction: The need for a simplified AI system to diagnose multiple diseases.

## 9. Discussion

A total of 178 studies were reviewed for this survey. All of the studies discussed mention work in DR screening systems using artificial intelligence techniques. There is a substantial need for automated, reliable systems for DR screening due to a substantial increase in DR patients worldwide. Studies considered publications published between January 2014 and June 2022. This study discusses various AI tools for DR followed by ML and DL techniques in DR. The numerous studies that created their own CNN framework versus others that usually use existing structures with transfer learning, such as VGG, ResNet, or AlexNet, are slightly different. Creating a CNN from scratch takes time and resources; however, employing transfer learning makes it easier and faster. The overall performance of its own CNN architecture is greater than that of the systems that used existing structures. This point should be taken into consideration by researchers, and further research should be conducted to differentiate between the two tendencies. Implicit and explicit feature extraction techniques are discussed, which help develop a model with reduced feature vector size and less machine effort, leading to better performance and speed. Publicly available datasets are discussed in detail with different properties. Performance measures serve as a matrix to evaluate the quality and accuracy of the research. They play a vital role in determining whether the research is desired. The availability of a robust DR detection technique capable of detecting all sorts of lesions and DR stages leads to a better follow-up strategy for DR patients, thereby avoiding loss of vision. This lack of technologies that might predict the five DR levels and detect DR lesions was a gap that needed to be filled. The above point could be viewed as a contemporary research question for researchers to pursue.

## 10. Conclusions

Automated methods for DR identification play a significant role in the early diagnosis of DR. A detailed review has been carried out, including 178 research studies found in Scopus, WOS, ophthalmology journals, Jama, PubMed, etc., to find the primary studies using the PRISMA approach. This review critically focuses on publicly available datasets, classification techniques used in ML and DL, and various traditional and currently used feature extraction methods, followed by various performance metrics used in DR. Traditional and novel biomarkers used in DR are highlighted. This study discovered and reported on several publicly available datasets with distinctive properties. In ML-based techniques, better performance is given by statistical-based characteristics also followed by shape and structure. ANN gives better performance for classification over SVM and, in the case of ML techniques, ensemble classifiers perform better. When this concerns DL, CNN was primarily applied to automatically extract and categorize the DR images. Accuracy, sensitivity, specificity, and area under the curve are the widely used performance metrics in DR. This review also described four novel research challenges in the DR detection field. This comprehensive review provides a profound overview of the topic of DR detection approaches and helpful insights to researchers working in this field. The scope of the evaluation can be expanded in the future to overcome limitations. Concepts such as transfer learning, ensemble learning, explainable AI, multi-task learning, and domain adaptation can be widely used in the future to detect DR at its early stages. Intelligent health monitoring technologies decrease the time to detect diagnoses, sparing ophthalmologists' time and cost, and allowing patients to communicate more quickly. The authors strongly believe that scientists and medical practitioners working in DR detection would benefit from this review. The readers of this article will gain the desired knowledge and obtain future research directions to extend their research work.

**Author Contributions:** Conceptualization, P.B. and S.G.; methodology, P.B., S.G. and K.K.; writing—original draft preparation, P.B.; writing—review and editing, P.B., S.G., K.P. and K.K.; visualization, S.G.; supervision, S.G., K.P. and K.K. All authors have read and agreed to the published version of the manuscript.

**Funding:** This research received no external funding.

**Institutional Review Board Statement:** Not applicable.

**Informed Consent Statement:** Not applicable.

**Data Availability Statement:** Not applicable.

**Conflicts of Interest:** The authors declare no conflict of interest.

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
