# Peer review of "A Systematic Literature Review on Diabetic Retinopathy Using an Artificial Intelligence Approach"

_2504-2289, doi:10.3390/bdcc6040152_

Round 1

Reviewer 1 Report

This paper studies A Comprehensive Survey on Diabetic Retinopathy Using 2 Artificial Intelligence ApproachThis is an interesting paper that could be a potentially publishable subject. There are a few weaknesses that should be addressed in this paper. Therefore, I suggest the authors resubmit it after a minor revision. My suggestions are as follows:

1. Discuss more the limitations of the study and future research suggestions if there are any.

2. The paper should be revised to include more recent references in 2021-2022.

3.  The quality of English needs to be improved across the paper. Also, the scientific terms pertinent to your topic should be improved.

4. Please explain the structure of your paper at the end of the introduction.

5.   Please explain more about “Evolution of DR using AI ” in line 145.

6. Please improve the quality of figure 14. It is a picture.

Reviewer 2 Report

This is an interesting manuscript representing a survey of AI applications in Diabetic Retinopathy (DR). The manuscript has a good introductory presentation of D.

A few comments that may help to improve the manuscript

1.       Please spell acronyms the first time that are presented (starting from the introduction) for example ML and DL

2.       The figures are not very professional for example in figure 1 the box lines are sometimes double and sometimes single. Figure 7 is blurry, perhaps the authors can do something to improve quality?

3.       There are some linguistic and punctuation issues for example double dots instead of single (line 98)

4.       There are pioneering works (since 1996) that the authors have not mentioned, see for example DOIs: 10.1136/bjo.80.11.940, and other papers around 2010: 10.3233/THC-130759, thus figure 6 is not very accurate and does not cover the history of AI in diabetic retinopathy since its beginning…. the authors may consider revising this figure and the provide the credit to such  pioneer researchers.

5.       Nice to see that authors have correctly followed the PRISMA approach in this study. Perhaps they could also reflect this in the title by mentioning that it is a systematic review?

Otherwise, in my opinion this manuscript is in general well written and is interesting for the journal audience.
